# Accelerating Matroid Optimization through Fast Imprecise Oracles

**Franziska Eberle**
Technical University of Berlin
Germany
f.eberle@tu-berlin.de

**Felix Hommelsheim**
University of Bremen
Germany
fhommels@uni-bremen.de

**Alexander Lindermayr**
University of Bremen
Germany
linderal@uni-bremen.de

**Zhenwei Liu**
University of Bremen
Germany
zhenwei@uni-bremen.de

**Nicole Megow**
University of Bremen
Germany
nmegow@uni-bremen.de

**Jens Schlöter**
Centrum Wiskunde & Informatica
The Netherlands
jens.schloter@cwi.nl

## Abstract

Querying complex models for precise information (e.g. traffic models, database systems, large ML models) often entails intense computations and results in long response times. Thus, weaker models that give imprecise results quickly can be advantageous, provided inaccuracies can be resolved using few queries to a stronger model. In the fundamental problem of computing a maximum-weight basis of a matroid, a well-known generalization of many combinatorial optimization problems, algorithms have access to a *clean* oracle to query matroid information. We additionally equip algorithms with a fast but *dirty* oracle. We design and analyze practical algorithms that only use few clean queries w.r.t. the quality of the dirty oracle, while maintaining robustness against arbitrarily poor dirty oracles, approaching the performance of classic algorithms for the given problem. Notably, we prove that our algorithms are, in many respects, best-possible. Further, we outline extensions to other matroid oracle types, non-free dirty oracles and other matroid problems.

## 1 Introduction

We study the power of a *two-oracle* model [1, 4, 5, 34, 36] for fundamental *matroid* optimization problems, which generalize many problems in combinatorial optimization.

The two-oracle model is an emerging technique for augmenting a problem where algorithms access information via *oracles*. The idea is to abstract from subroutines, such as Neuronal Network (NN) inference or graph algorithms, which compute required information from underlying models. Already today the size of such models can be arbitrarily large, and they are expected to grow further in the near future. Thus, computing *precise* results for each oracle query can be (too) expensive. To mitigate these costs, we assume to have access to a second oracle that is *less expensive* but gives *possibly imprecise* answers. Such an oracle can be an efficient heuristic or a smaller NN. The goal is to leverage this fast *dirty* oracle to obtain enough information in order to query the expensive *clean* oracle as little as possible. This model has been successfully applied to, e.g., clustering [4, 34], sorting [1], priority queues [5], or data labeling [36].

We study this model in the context of matroid optimization. Matroids play a central and unifying role in combinatorial optimization as numerous classic problems can be framed as matroid basis problems, e.g., problems in resource allocation and network design. A *matroid* $\mathcal{M} = (E, \mathcal{I})$ is a

38th Conference on Neural Information Processing Systems (NeurIPS 2024).

downward-closed set system with a certain augmentation property. It is represented by a ground set $E$ of elements and a family $\mathcal{I} \subseteq 2^E$ of *independent* sets. A *basis* is an inclusion-wise maximal independent set, and all bases of a matroid have the same size, which is the *rank* of the matroid; we give formal definitions later. A prominent matroid is the *graphic* matroid in which, given an undirected graph, every subset of edges that induces an acyclic subgraph is independent.

Since $|\mathcal{I}|$ can be exponential in $n := |E|$, algorithms operating on matroids are given access to oracles. The *independence oracle* answers for a query $S \subseteq E$ whether $S \in \mathcal{I}$. Given a weight $w_e \geq 0$ for every $e \in E$, a classic goal in matroid optimization is to find a basis of $\mathcal{M}$ of maximum total weight. In his seminal work, Edmonds [18] showed that the *greedy algorithm*, which greedily adds elements in order of non-increasing weight, solves this problem using an optimal number of $n$ oracle calls, and, vice versa, that matroids are the most general downward-closed set systems for which this algorithm is correct. For graphic matroids, this greedy algorithm corresponds to the classic algorithm of Kruskal [15] for computing a minimum-weight spanning tree in an edge-weighted graph, which is commonly taught in undergraduate algorithm classes.

To motivate the two-oracle model for matroid optimization, we continue with the special case of computing a minimum spanning tree in a large network. This problem arises, e.g., to ensure connectivity in a telecommunication network or as a subroutine to approximate a cheap tour between certain points of interest in a road network. Here, the elements $E$ correspond to the edges of the network. The clean oracle for this graphic matroid has to decide whether a set of edges is cycle-free. While this is doable in time linear in the query size, it can be prohibitively expensive for huge queries. However, there are several ways to design a dirty oracle of reasonable quality:

- Networks, especially telecommunication networks, often evolve over time. In this case, the dirty oracle could quickly return cached results of the previous (now outdated) network.

- If the network is clustered into many highly connected components that are only loosely connected to each other, such as road networks are clustered around big cities, the dirty oracle can check the query for each (city) component individually. Since there may only be few highways between the cities, this can be quite close to the clean answer.

- The dirty oracle could operate in a sparsified subnetwork, e.g., by restricting it to only major roads or data links.

We initiate the study of the following two-oracle model for general matroid problems: In addition to the clean oracle of a given matroid $\mathcal{M} = (E, \mathcal{I})$, an algorithm has access to a fast dirty oracle. For simplicity, we assume that this oracle belongs to a matroid $\mathcal{M}_d = (E, \mathcal{I}_d)$ and answers for a query $S$ whether $S \in \mathcal{I}_d$. In fact, the former assumption is not necessary, and we will adapt our results to weaker dirty-oracle variants such as arbitrary downward-closed set systems. We emphasize that the algorithm has no knowledge about the relationship between the dirty oracle and the true matroid $\mathcal{M}$. Finally, we assume for now that calling the dirty oracle is free and discuss mitigations later. Since calling the dirty oracle is free, our goal is to minimize the number of clean-oracle calls required to solve the underlying matroid problem.

Our work neatly aligns with the celebrated framework of *learning-augmented algorithms* [30], which is receiving tremendous attention [29]. Here, an algorithm has access to a problem-specific *prediction* and yields a good performance when the prediction is accurate (*consistency*) and does not perform too bad when given arbitrary predictions (*robustness*). Ideally, the performance degrades gracefully with the prediction's quality, measured by an *error* function (*smoothness*). The type of performance thereby depends on the problem, and could be, e.g., a quality guarantee of the computed solution of an optimization problem, or the running time of an algorithm. The latter is intricately related to the goal of the two-oracle model: the overall running time highly depends on the number of clean-oracle calls. From this perspective, the *consistency* of an algorithm in the two-oracle model is the worst-case number of clean-oracle calls it uses when the dirty oracle is perfect, i.e., $\mathcal{M} = \mathcal{M}_d$, while the *robustness* is the worst-case number of clean oracles independent of the quality of the dirty oracle.

More generally, we can interpret algorithms for one model also in the other model and vice versa: On the one hand, our two-oracle model fits into the learning-augmented algorithm framework by considering an optimal solution w.r.t. the dirty oracle as a possibly imprecisely predicted solution. On the other hand, a predicted solution $B_d \subseteq E$ can be used to construct the dirty oracle $\mathcal{M}_d = (E, 2^{B_d})$. Thus, our main results are also applicable in the learning-augmented setting.

## 1.1 Our results

In this paper, we design optimal algorithms in the two-oracle model for the problem of finding a maximum-weight basis (defined later) of a matroid.

1. **Two-oracle algorithms:** For any integer $k \geq 1$, there is an algorithm which computes a maximum-weight basis of a matroid $\mathcal{M}$ using at most

$$\min\left\{n - r + k + \eta_A \cdot (k+1) + \eta_R \cdot (k+1)\lceil \log_2 r_d \rceil, \left(1 + \frac{1}{k}\right)n\right\}$$

   many oracle calls to $\mathcal{M}$ (see Theorem 3.8).

Here, $r$ is the rank of the matroid, i.e., the size of any basis, and $r_d$ the rank of $\mathcal{M}_d$. In terms of learning-augmented algorithms, our algorithm has a consistency of at most $n - r + k$ and a robustness of at most $(1 + \frac{1}{k})n$. Moreover, the bound of our algorithm smoothly degrades w.r.t. the quality of the dirty oracle, measured by our error measures $\eta_A$ and $\eta_R$ (the *prediction errors*). Intuitively, $\eta_A$ and $\eta_R$ denote the number of elements that have to be added to and removed from a maximum-weight basis of $\mathcal{M}_d$, respectively, to reach a maximum-weight clean basis; we give a formal definition later. Our algorithm has a tuning parameter $k$ to regulate the level of robustness against a dirty oracle of bad quality; a beneficial feature when the quality of the dirty oracle can be roughly estimated. In Section 2, we present slightly improved bounds for the case of unit weights.

Observe that, given a dirty oracle of reasonable quality, our algorithm significantly improves upon the greedy algorithm whenever the rank $r$ is not too small (due to the dependence on $n - r$), which is always the case for, e.g., graphic matroids in sparse networks like road or telecommunication networks. We further show that this dependence is best possible for any deterministic algorithm for *any* rank $r$. Moreover, our algorithm is optimal w.r.t. to the dependence on $\eta_A$ and $\eta_R$:

2. **Tight lower bounds:** For every rank $r$, every deterministic algorithm requires at least $n - r + \eta_A$ and at least $n - r + \eta_R \lceil \log_2 r_d \rceil$ clean-oracle calls to compute a basis of a matroid (Appendix B).

Yet we present an algorithm which bypasses the dependence on $n - r$ by leveraging the slightly more powerful clean rank oracle (see Section 4.1 for the definition). Note that any algorithm requires at least $n$ clean rank oracle calls in the traditional setting for this problem in the worst case.

3. **Rank oracles:** There is an algorithm which computes a basis of a matroid $\mathcal{M}$ using at most $2 + \eta_A \lceil \log_2(n - r_d) \rceil + \eta_R \lceil \log_2 r_d \rceil$ and at most $n + 1$ calls to the rank oracle of $\mathcal{M}$ (Section 4.1).

Finally, we initiate the study of extensions, with which we hope to foster future research.

4. **Costly oracles:** In this model, every dirty-oracle call incurs cost 1, and every clean-oracle call costs $p > 1$. We are interested in minimizing the total cost. We illustrate that this model requires new algorithmic techniques compared to our main setting (see Section 4.2).

5. **Matroid intersection:** We give two different approaches on how our techniques can be incorporated in textbook algorithms for reducing the number of clean-oracle calls in the fundamental matroid intersection problem using dirty oracles (see Section 4.3).

All omitted proofs are deferred to the appendix.

## 1.2 Further related work

**Noisy oracles, two-oracle model, imprecise predictions.** Optimization in the presence of imprecise oracles is a fundamental problem and has been studied extensively, also for submodular optimization [25, 26, 27, 28, 35], which is connected to matroid optimization via the submodularity of rank functions. The majority of previous work assumes access only to a *single noisy* oracle, where the noise usually is of stochastic nature. Only recently, a two-oracle model as in our work has been studied from a theoretical perspective. Bai and Coester [1] consider sorting with a clean and a dirty comparison operator. They minimize the number of clean-comparison-operator calls and give guarantees that smoothly degrade with the number of wrong dirty-comparison answers. Similar strong and weak oracles have also been considered by Bateni et al. [4] for the Minimum Spanning

Tree problem and clustering, and by Benomar and Coester [5] in the context of priority queues. In contrast to our model, they consider oracles for accessing the distance between two points and not for deciding cycle freeness (graphic matroid). Besides these explicit results on two-oracle models, explorable uncertainty with predictions [20, 21] can also be interpreted as such a model.

**Two- and multistage problems.** All algorithms presented in this paper also solve the following two-stage problem: Given a maximum-weight basis $B_d$ for a first-stage matroid $\mathcal{M}_d$ (dirty matroid), compute a maximum-weight basis $B$ for a second stage matroid $\mathcal{M}$ (clean matroid) with minimum $|B_d \triangle B|$, where $B_d \triangle B$ denotes the symmetric difference of $B_d$ and $B$, and minimize the number of oracle calls to $\mathcal{M}$. Two- (or multi-) stage problems of this type have been studied extensively, mostly for graph problems [2, 3, 12, 13, 19, 22, 23, 33] but also for matroids [8, 9, 14, 24]. Most of these works consider a combined objective optimizing the quality of the second stage solution *and* the distance between the first- and second-stage solutions. In contrast, we insist on an optimal second-stage solution and minimize the number of clean-oracle calls. Furthermore, to our knowledge, all previous work on matroid problems in these models assumes that the matroid stays the same for all stages but the weights of the elements change, whereas we assume the opposite. Blikstad et al. [7] consider the somewhat similar problem of dynamically maintaining a basis of a matroid, but in a different oracle model.

### 1.3 Preliminaries

**Matroids.** A *matroid* $\mathcal{M}$ is a tuple $(E, \mathcal{I})$ consisting of a finite ground set $E$ of $n$ elements and a family of *independent sets* $\mathcal{I} \subseteq 2^E$ with $\emptyset \in \mathcal{I}$ that satisfy the following properties: (i) $\mathcal{I}$ is downward-closed, i.e., $A \in \mathcal{I}$ implies $B \in \mathcal{I}$ for all $B \subseteq A$ and (ii) if $A, B \in \mathcal{I}$ with $|A| > |B|$, then there exists $a \in A \setminus B$ s.t. $B + a \in \mathcal{I}$. (We write $X + e$ when we add $e \in E \setminus X$ to $X \subseteq E$ and $X - e$ when we remove $e \in X$ from $X \subseteq E$.) An important notion are *bases*, which are the (inclusion-wise) maximal elements of $\mathcal{I}$; for a fixed matroid, we denote the set of bases by $\mathcal{B}$. For a dirty matroid $\mathcal{M}_d$, we refer to the set of bases by $\mathcal{B}_d$. A *circuit* is a minimal dependent set of elements. The main results of this paper consider the problem of finding a maximum-weight basis, i.e., given matroid $\mathcal{M} = (E, \mathcal{I})$ and weights $w_e$ for all $e \in E$, the goal is to find a basis $B \in \mathcal{B}$ maximizing $\sum_{e \in B} w_e$. The *greedy algorithm* solves this problem by iteratively adding elements in non-increasing order of their weight to the solution if possible, i.e., if the solution stays independent in $\mathcal{M}$ [18]. Given a weighted ground set, we always assume that the elements of $E = \{e_1, \ldots, e_n\}$ are indexed according to the weight order, i.e., $i \leq j$ implies $w_{e_i} \geq w_{e_j}$, with ties broken arbitrarily. Given that, for any $i$ and $S \subseteq E$ we define $S_{\leq i} = \{e_j \in S \mid j \leq i\}$ ($S_{\geq i}$ analogously).

**Requirements on algorithms and optimal solutions.** For all considered problems, we require algorithms to execute clean queries (oracle calls) until the answers to these queries reveal sufficient information to solve the given problem, e.g., find a maximum-weight basis for the clean matroid. More precisely, the queries executed by the algorithm together with the answers must be a *certificate* that a third party with only access to the clean matroid and without any additional knowledge can use in order to find a provable solution. In particular, an optimal algorithm that knows the answers to all clean queries upfront has to execute queries in order to satisfy the certificate requirement.

We refer to the *robustness* of an algorithm, when bounding the maximum number of clean-oracle calls the algorithm needs for any input instance, independently of the quality of the dirty oracle.

**Definition of our error measure.** We define an error measure that quantifies the quality of the dirty oracle w.r.t. the clean oracle. We define the error measure for the case that the dirty oracle is a matroid and describe in the next section how this extends to arbitrary downward-closed set systems. Let $\mathcal{B}^*$ be the set of maximum-weight bases of $\mathcal{M}$ and $\mathcal{B}_d^*$ be the set of maximum-weight bases of $\mathcal{M}_d$. (In the unweighted case, $\mathcal{B}^* = \mathcal{B}$ and $\mathcal{B}_d^* = \mathcal{B}_d$.) We first define for every $S \in \mathcal{B}_d^*$ the sets $A(S), R(S)$ as any cardinality-wise smallest set $A \subseteq E \setminus S$ and $R \subseteq S$, respectively, such that $S \cup A \supseteq B$ and $S \setminus R \subseteq B'$ for some $B, B' \in \mathcal{B}^*$.

These sets describe the smallest number of additions/removals necessary to transform $S$ into a superset/subset of some maximum-weight basis of $\mathcal{M}$. We call $|A(S)| + |R(S)|$ the *modification distance* from $S$ to $\mathcal{B}^*$. Our final error measure is defined as the largest modification distance of any maximum-weight basis of $\mathcal{M}_d$, that is, $\eta_A = \max_{S \in \mathcal{B}_d^*} |A(S)|$ and $\eta_R = \max_{S \in \mathcal{B}_d^*} |R(S)|$.

Assume both oracles are matroids. It follows from standard matroid properties that, for any dirty basis $B_d$, there are modification sets $A, R$ with $|A| = |A(B_d)|, |R| = |R(B_d)|$ and $B_d \setminus R \cup A \in \mathcal{B}$.

Hence $r = r_d + \eta_A - \eta_R$. Also, for all $S_1, S_2 \in \mathcal{B}_d$, $|A(S_1)| \leq |A(S_2)|$ if and only if $|R(S_1)| \leq |R(S_2)|$.

## 1.4 Discussion of the model

**Computing a dirty basis upfront.** For all our results on computing a (maximum-weight) basis, it suffices to first compute a (maximum-weight) dirty basis and afterwards switch to exclusively using the clean oracle. A similar observation can be made for the results of Bai and Coester [1] on sorting, where it would be possible to initially compute a tournament graph for the dirty comparator and afterwards switch to only using the clean comparator without increasing the number of clean-comparator calls. In Section 4, we observe that separating the usage of the dirty and clean oracles in this way does not work anymore if dirty-oracle calls also incur costs.

**Counting wrong answers as error measure.** Bai and Coester [1] use the number of wrong dirty-comparator answers as error measure. While this is a meaningful error measure for sorting, a similar error measure for our problem does not seem to accurately capture the quality of the dirty oracle. Consider an instance with unit weights, where we have $\mathcal{B}_d \subset \mathcal{B}$. This can lead to an exponential number of wrong dirty oracle answers, but the dirty oracle still allows us to compute a clean basis. For this reason, we use the modification distance as error measure instead.

**Relaxing requirements on the dirty oracle.** We illustrate how to extend the error definition of Section 1.3 to arbitrary downward-closed set systems in the unweighted case: For every *inclusion-wise maximal set* $S$ of the dirty oracle we compute $A(S)$ and $R(S)$ as before. The addition and removal error are then defined analogously by replacing $\mathcal{B}_d^*$ with the set of all inclusion-wise maximal independent sets (instead of all maximum sets in the unweighted case). Then, all our results in this paper also carry over to this more general setting. Using an error definition with respect to some greedy algorithm on weighted downward-closed set systems, one can also obtain the same result for arbitrary *weighted* downward-closed set systems. However, for clarity, we only prove our statements for the case that the dirty oracle is a matroid.

## 1.5 Organization of the paper

We begin in Section 2 with a gentle introduction to our techniques and algorithms by studying the simpler problem of computing any basis of a matroid. Then, we extend these ideas in Section 3 and present our main algorithmic results. Finally, in Section 4, we demonstrate how to generalize and adapt our approach to other settings and problems. Many proofs for these sections are deferred to the appendix. In Appendix B, we also included a detailed section on our lower bounds.

## 2 Warm-up: computing an unweighted basis

The goal of this section is to give a gentle introduction to our algorithmic methods, which we extend in the next sections. Consider the basic problem of finding any basis of a matroid. Note that without the dirty oracle, we need exactly $n$ clean-oracle calls to find and verify any basis in the worst-case.

In the following, we assume that we are given an arbitrary basis $B_d$ of the dirty matroid $\mathcal{M}_d$, which can be computed without any call to the clean oracle. We first analyze the so-called *simple algorithm*: If $B_d \in \mathcal{I}$, we set $B$ to $B_d$. Otherwise, we set $B$ to the empty set. Next, for each element $e \in E \setminus B$, we add it to $B$ if $B + e \in \mathcal{I}$ and output $B$ at the end.

The idea of the simple algorithm is to only use the dirty basis $B_d$ if it is independent in the clean matroid, as we then can easily augment it to a clean basis. Otherwise, we abandon it and effectively run the classic greedy algorithm. Formalizing this sketch proves the following lemma.

**Lemma 2.1.** *The simple algorithm computes a clean basis using at most $n + 1$ clean-oracle calls. Further, if $B_d \in \mathcal{B}$, it terminates using at most $n - r + 1$ clean-oracle calls.*

Surprisingly, in Appendix B, we will see that this simple algorithm achieves a best-possible trade-off between optimizing the cases $B_d \in \mathcal{B}$ and $B_d \notin \mathcal{B}$ at the same time.

## 2.1 An error-dependent algorithm

The simple algorithm discards the dirty basis $B_d$ if it is not independent in $\mathcal{M}$. This approach may be wasteful, especially if removing just one element from $B_d$, such as in the case of a circuit, leads to independence. This seems particularly drastic if the clean and dirty oracle are relatively "close", i.e., the error measured by the modification distance is small. This suggests a refinement with a more careful treatment of the dirty basis $B_d$. We propose a binary search strategy to remove elements from $B_d$ until it becomes independent. A key feature is that this allows for an error-dependent performance guarantee bounding the number of clean-oracle calls by $\eta_A$ and $\eta_R$, the smallest numbers of elements to be added and removed to turn a dirty basis into a basis of the clean matroid.

We define the *error-dependent algorithm*: First, set $B$ to $B_d$ and fix an order of the elements in $B$. Repeatedly use binary search to find the smallest index $i$ s.t. $B_{\leq i} \notin \mathcal{I}$ and remove $e_i$ from $B$ until $B \in \mathcal{I}$. Then add each element $e \in E \setminus B_d$ to $B$ if $B + e \in \mathcal{I}$ and output the final set $B$.

**Lemma 2.2.** *The error-dependent algorithm computes a clean basis using at most $n - r + 1 + \eta_A + \eta_R \cdot \lceil \log_2 r_d \rceil$ clean-oracle calls.*

*Proof.* The algorithm simulates finding a maximal independent subset of $B_d$ w.r.t. the clean matroid, and augments the resulting set to a clean basis. Hence, the correctness follows from matroid properties. We remove $|R(B_d)| \leq \eta_R$ elements from $B_d$. Hence, the removal loop is executed $|R(B_d)|$ times. In each iteration, we use at most $1 + \lceil \log_2 r_d \rceil$ clean queries (one for checking $B \in \mathcal{I}$ and at most $\lceil \log_2 r_d \rceil$ for the binary search). Thus, the removing process uses at most $|R(B_d)|(1 + \lceil \log_2 r_d \rceil)$ clean queries. Augmenting $B$ uses $n - r_d$ clean queries. Combined, our algorithm uses at most $|R(B_d)|(1 + \lceil \log_2 r_d \rceil) + n - r_d + 1$ oracle calls. We conclude using $|R(B_d)| \leq \eta_R$ and $r = r_d - \eta_R + \eta_A$. $\qquad\square$

## 2.2 An error-dependent and robust algorithm

The error-dependent algorithm has a good bound on the number of clean-oracle calls (less than $n$) when $\eta_A$ and $\eta_R$ are small. However, in terms of *robustness*—i.e., the maximum number of oracle calls for any instance, regardless of the dirty-oracle quality—this algorithm performs asymptotically worse than the gold standard $n$, achieved by the classic greedy algorithm. This is the case when the dirty basis is equal to $E$, but the clean matroid is empty: the error-dependent algorithm executes $n$ binary searches over narrowing intervals, using $\log_2(n!) \in \Theta(n \log n)$ clean-oracle calls. By looking closer at the error-dependent algorithm, the special structure of this example can be explained because queries charged to $n - r + \eta_A$ are essentially also done by the greedy algorithm. Hence, they are in a sense already robust. Motivated by the greedy algorithm, another extreme variant of the removal process would be to go linearly through $B_d$ and greedily remove elements. This gives an optimal robustness, but is clearly bad if $\eta_R$ is small.

For our main result in the unweighted setting, we combine both extremes into a robustification framework and achieve a trade-off using the following key observation. If we have to remove many elements ($\eta_R$ is large), some elements must be close to each other. In particular, if the next removal is close to the last one (in terms of the fixed total order), a linear search costs less than a binary search. Based on this, after a removal, we first check the next $\Theta(\log(r_d))$ elements of $B_d$ linearly for other removals before executing a binary search. This bounds the number of binary searches by $\Theta(\frac{r_d}{\log(r_d)})$, each incurring a cost of $\lceil \log_2(r_d) \rceil$. However, the linear search also incurs some cost. Thus, we further parameterize this idea (see Algorithm 3), and obtain the following main result.

**Theorem 2.3.** *For every $k \in \mathbb{N}_+$, there is an algorithm that, given a dirty matroid $\mathcal{M}_d$ of rank $r_d$ with unknown $\eta_A$ and $\eta_R$, computes a basis of a matroid $\mathcal{M}$ of rank $r$ with at most $\min\{n - r + k + \eta_A + \eta_R \cdot (k+1)\lceil \log_2 r_d \rceil, (1 + \frac{1}{k})n\}$ oracle calls to $\mathcal{M}$.*

## 3 Computing a maximum-weight basis

Consider the weighted setting. Recall that $\mathcal{B}_d^*$ and $\mathcal{B}^*$ denote the sets of maximum-weight bases of the dirty and clean matroid, respectively. We assume that we are given a maximum-weight basis $B_d \in \mathcal{B}_d^*$, which can be computed without any clean-oracle calls. The main difficulty compared to the unweighted setting is as follows: In the unweighted setting, the error-dependent algorithm first

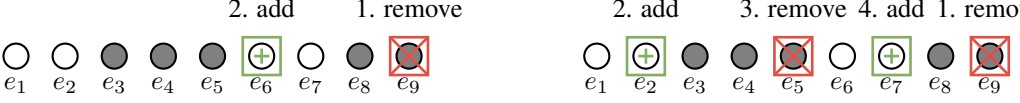

(a) Modification to an arbitrary clean basis.

(b) Modification to a maximum-weight clean basis. Adding $e_2$ is necessary for its high weight. Element $e_5$ is only blocking after adding $e_2$ to $B_d - e_9$, hence cannot be detected earlier.

Figure 1: A matroid with elements $e_1, \ldots, e_9$ (displayed as circles) ordered left-to-right by non-increasing weight. The elements of the maximum-weight dirty basis $B_d$ are filled.

computes an *arbitrary* maximal independent set $B' \subseteq B_d$ and then easily augments $B'$ to a clean basis. In the weighted setting, however, there clearly can be independent sets $B' \subseteq B_d$ that are not part of any maximum-weight basis; hence we need to be more careful. Finding such a special independent subset of $B_d$ only by removing elements from $B_d$ and testing its independence seems difficult: $B_d$ itself could be independent, but not part of any maximum-weight basis. However, even in this case, $B_d$ can be very close to a maximum-weight basis w.r.t. $\eta_A$ and $\eta_R$. Therefore, we cannot avoid carefully modifying $B_d$ since strategies like the greedy algorithm would use too many queries.

Thus, we alternatingly add and remove elements to and from $B_d$. Intuitively, we want to ensure, as the greedy algorithm, that we do not miss adding elements with large weight. Thus, we try to add them as soon as possible. However, even if they are part of every basis in $\mathcal{B}^*$, this might result in a *dependent* current solution unless we now remove elements, which were not detectable before. An example of such a situation is given in Figure 1. This observation rules out simple two-stage algorithms as used in the unweighted case.

We now present our algorithm. Its full description is shown in Algorithm 1. Given elements $E = \{e_1, \ldots, e_n\}$ in non-decreasing order of their weight, it maintains a preliminary solution, which initially is set to the dirty basis $B_d$. It modifies this solution over $n$ iterations and tracks modifications in the variable sets $A \subseteq E \setminus B_d$ (*added* elements) and $R \subseteq B_d$ (*removed* elements). In every iteration $\ell$, the algorithm selects elements to add and remove such that *at the end* of iteration $\ell$ its preliminary solution $(B_d \setminus R) \cup A$ satisfies two properties, which we define and motivate now.

Property $(i)$ requires that the preliminary solution $(B_d \setminus R) \cup A$ up to $e_\ell$ should be a *maximal* subset of some maximum-weight basis. For the sake of convenience, we introduce the matroid $\mathcal{M}^* = (E, \mathcal{I}^*)$, where $\mathcal{I}^*$ is the set of all subsets of $\mathcal{B}^*$. Then, we can use the following definition.

**Definition 3.1.** A set $S$ is $k$-*safe* if $S_{\leq k} \in \mathcal{I}^*$ and for every $e \in E_{\leq k} \setminus S_{\leq k}$ it holds that $S_{\leq k} + e \notin \mathcal{I}^*$.

In other words, a set $S$ is $k$-safe if it is a basis of the truncated matroid of the $k$th prefix of $E$. Using this definition, Property $(i)$ requires that at the end of iteration $\ell$, the current solution $(B_d \setminus R) \cup A$ is $\ell$-safe. Establishing this property in every iteration will give us that the final solution is $n$-safe, and, thus, a maximum-weight basis of $\mathcal{M}$. This works because the algorithm does not modify its solution for the $\ell$th prefix of $E$ after the $\ell$th iteration.

To establish Property $(i)$ after every iteration without using too many clean queries, it also maintains Property $(ii)$: at the end of every iteration the current solution is independent, i.e., $(B_d \setminus R) \cup A \in \mathcal{I}$.

We now give some intuition on how our algorithm achieves this. Initially, say for $\ell = 0$, before Line 2 Property $(i)$ is fulfilled trivially. For Property $(ii)$, before Line 5 the algorithm greedily adds minimum-weight elements of $B_d$ to $R$ that close a circuit in the smallest prefix of $(B_d \setminus R) \cup A$ (Lines 2-4). This subroutine can be implemented via binary search. To guarantee both properties at the end of iteration $\ell$, first observe that Property $(ii)$ holds as long as $A$ and $R$ have not changed in this iteration. However, this might be necessary to establish Property $(i)$. Intuitively, our algorithm wants to act like the classic greedy algorithm to ensure $\ell$-safeness. Thus, it checks whether $e_\ell$ should be in the solution by considering its solution for $E_{\leq \ell}$. Clearly, if $e_\ell \in B_d \setminus R$ or $e_\ell \in R$, there is nothing to modify (Line 6), because the current solution is independent due to Property $(ii)$ for the previous iteration. Similarly, if $e_\ell \notin B_d$ and the current solution for $E_{\leq \ell}$ together with $e_\ell$ is independent, we add $e_\ell$ to our solution (Lines 6-7). However, by adding an element, Property $(ii)$ can become false due to an introduced circuit, which we have to break (Lines 8-10). Finally, there might be the situation that $e_\ell \in B_d$ has been added to $R$ in an earlier iteration, so $e_\ell \in R$. In this

---

**Algorithm 1:** Find a maximum-weight basis

---

**Input:** dirty basis $B_d \subseteq E$, matroid $\mathcal{M} = (E, \mathcal{I})$

1   $A \leftarrow \emptyset; R \leftarrow \emptyset$ {*added / removed* elements}

2   **while** $(B_d \setminus R) \cup A \notin \mathcal{I}$ **do**

3      Find the *smallest* index $i$ s.t. $((B_d \setminus R) \cup A)_{\leq i} \notin \mathcal{I}$ via *binary search*

4      $R \leftarrow R + e_i$

5   **for** $\ell = 1$ *to* $n$ **do**

6      **if** $e_\ell \notin B_d$ *and* $((B_d \setminus R) \cup A + e_\ell)_{\leq \ell} \in \mathcal{I}$ **then**

7         $A \leftarrow A + e_\ell$

8         **if** $(B_d \setminus R) \cup A \notin \mathcal{I}$ **then**

9            Find the *smallest* index $i$ s.t. $((B_d \setminus R) \cup A)_{\leq i} \notin \mathcal{I}$ via *binary search*

10            $R \leftarrow R + e_i$

11 **return** $(B_d \setminus R) \cup A$

---

case, we clearly do not want to even consider adding $e_\ell$ again, as queries for verifying this addition cannot be bounded by our error measure. Indeed, removing and re-adding an element cannot be part of any minimal modification distance. Thus, our algorithm skips such elements (Line 6). To justify this, we prove that removing element $e_\ell$ is always necessary, in the sense that there always is a circuit that $e_\ell$ breaks and that cannot be broken by removing elements in *later* iterations by Lines 8-10. This follows from classic matroid properties for circuits. Formally, we prove in Appendix C:

**Lemma 3.2** (Property (ii)). *At the start (end) of each iteration of Line 5, it holds* $(B_d \setminus R) \cup A \in \mathcal{I}$.

**Lemma 3.3** (Property (i)). *At the end of every iteration $\ell$ of Line 5, $(B_d \setminus R) \cup A$ is $\ell$-safe.*

Since there are at most $n$ elements in $R$, the algorithm clearly terminates, and we conclude as follows.

**Corollary 3.4.** *Algorithm 1 terminates with an $n$-safe set.*

It remains to bound the number of clean queries. Fix $A$ and $R$ to their final sets. Assume that elements are non-increasingly ordered by their weight; among elements of *equal weight*, elements of $B_d$ come *before* elements of $E \setminus B_d$. For such an ordering, the algorithm modifies $B_d$ to a closest basis of $\mathcal{B}^*$.

**Lemma 3.5.** *It holds that $|A| \leq \eta_A$ and $|R| \leq \eta_R$.*

To conclude, we use a charging scheme and Lemma 3.5 to derive the following bound; for details, we refer to Appendix C.

**Lemma 3.6.** *Algorithm 1 computes a max-weight basis with at most $n - r + 1 + 2\eta_A + \eta_R \cdot \lceil \log_2(r_d) \rceil$ clean queries.*

We complement this algorithmic result by a lower bound. It proves that our error-dependent guarantees in the unweighted case are not possible in the weighted case. Hence, it separates both settings.

**Lemma 3.7.** *Every deterministic algorithm for finding a maximum-weight basis executes strictly more than $n - r + \eta_A + \eta_R \cdot \lceil \log_2(r_d) \rceil + 1$ clean-oracle calls in the worst-case.*

**Application of the robustification framework.** As in the unweighted case, our algorithm may perform poorly when the dirty oracle is of low quality, i.e., $\eta_A$ and $\eta_R$ are large. We extend the ideas for robustifying the error-dependent algorithm (cf. Section 2.2) and combine them with the concepts developed above. The key idea for robustifying Algorithm 1 is to start a binary search only after a sufficient number of linear search steps. However, as observed above (cf. Figure 1), we cannot remove all blocking elements in one iteration. While the simple argument that the linear search partitions $B_d$ still holds, it does not cover the total removal cost, because a later addition can create a removal at a previously checked position. To overcome this, we observe that in Algorithm 1, immediate removal of every detected element from the current solution is not necessary; we just need to decide whether in iteration $\ell$ element $e_\ell$ should be part of the solution.

**Algorithm 2:** Find a maximum-weight basis (robustified)

---

**Input:** dirty basis $B_d$, matroid $(E, \mathcal{I})$, integer $k \geq 1$

1   $A \leftarrow \emptyset; R \leftarrow \emptyset; d_{\max} \leftarrow \max_{e_i \in B_d} i$

2   $q \leftarrow 0; \mathrm{LS} \leftarrow \mathrm{true}$ {linear search counter / flag}

3   **for** $\ell = 1$ *to* $n$ **do**

4      **if** $e_\ell \notin B_d$ **then**

5         **if** $((B_d \setminus R) \cup A + e_\ell)_{\leq \ell} \in \mathcal{I}$ **then** $A \leftarrow A + e_\ell$ and $\mathrm{LS} \leftarrow \mathrm{true}$

6      **else if** $e_\ell \in B_d \setminus R$ *and LS* $= \mathrm{true}$ **then**

7         $q \leftarrow q + 1$

8         **if** $((B_d \setminus R) \cup A)_{\leq \ell} \notin \mathcal{I}$ **then** $R \leftarrow R + e_\ell$ and $q \leftarrow 0$

9         **if** $\ell = d_{\max}$ *or* $(q = k - 1$ *and* $(B_d \setminus R) \cup A \in \mathcal{I})$ **then** $q \leftarrow 0$ and $\mathrm{LS} \leftarrow \mathrm{false}$

10        **else if** $q = k\lceil \log_2 r_d \rceil$ **then**

11           Find the *smallest* index $i$ s.t. $((B_d \setminus R) \cup A)_{\leq i} \notin \mathcal{I}$ via *binary search*

12           $R \leftarrow R + e_i$ and $q \leftarrow 0$

13   **return** $(B_d \setminus R) \cup A$

---

In our robustified algorithm (Algorithm 2) we exploit this as follows. While in Algorithm 1 we linearly check prefixes of $E \setminus B_d$ for additions, we now linearly check prefixes of $E$ for additions (Lines 4-5) *and* removals (Lines 6-8). However, for the sake of a good error-dependency, we count these removal checks (cf. increment counter $q$ in Line 7) and execute a binary search only if we checked enough elements in $B_d$ linearly (Lines 10-12). Then, we can again bound the total cost for the binary searches using a density argument. Whenever we remove an element, we charge the previous cost of the linear searches to this removal error and reset $q$, which re-activates the linear search. However, if the current solution is already independent, we do not want to search for removal errors at all (cf. Lines 2 and 8 in Algorithm 1). Unfortunately, doing such a check after every addition and removal already rules out a robustness close to $n$. Thus, we slightly delay this check w.r.t. the counter $q$, and stop the removal search accordingly (Line 9). Finally, whenever an element is added, we make sure that the linear search is running or that we start it again (Line 5), as a new circuit could have been introduced in our solution. Formalizing this sketch proves our main theorem.

**Theorem 3.8.** *For any $k \in \mathbb{N}_+$, there is an algorithm that, given a dirty matroid $\mathcal{M}_d$ of rank $r_d$ with unknown $\eta_A$ and $\eta_R$, computes a maximum-weight basis of a matroid $\mathcal{M}$ of rank $r$ with at most $\min\{n - r + k + \eta_A \cdot (k + 1) + \eta_R \cdot (k + 1)\lceil \log_2 r_d \rceil, (1 + \frac{1}{k})n\}$ oracle calls to $\mathcal{M}$.*

## 4 Extensions and future work

### 4.1 Rank oracles

Another common type of matroid oracles is the rank oracle: Given any $S \subseteq E$, a rank oracle returns the cardinality of a maximum independent set contained in $S$, denoted by $r(S)$. Since $r(S) = |S|$ if and only if $S \in \mathcal{I}$, our algorithmic results for independence oracles directly transfer. Moreover, for the unweighted setting, we can even reduce the number of oracle calls using a rank oracle, implying that some lower bounds do not translate. For example, given $B_d$ we can compute its rank $r(B_d)$ to obtain $\eta_R = |B_d| - r(B_d)$ and decide whether to remove elements via binary search or immediately switch to the greedy algorithm. Further, we can improve the dependency on $\eta_A$ if $\eta_A$ is small as we can find the elements to be added via a binary search. Hence, we get the following result.

**Proposition 4.1.** *There is an algorithm that computes a clean basis with at most $\min\{n + 1, 2 + \eta_R \cdot \lceil \log_2 r_d \rceil + \min\{\eta_A \cdot \lceil \log_2(n - r_d) \rceil, n - r_d\}\}$ clean rank-oracle calls.*

The full discussion on rank oracles can be found in the appendix. For future work it would be interesting to see if the error-dependency and the worst-case bound can be improved, and if rank oracles can be used to improve the results for the weighted setting.

## 4.2 Dirty independence oracle with cost

We consider the generalized setting where a dirty-oracle call has cost 1 and a clean-oracle call has cost $p > 1$ with the objective to minimize the total cost. Lemma B.1 translates to this setting, giving a lower bound $p(n - r + 1)$. Note that the previous results assume that $p \gg 1$ in this setup.

The main takeaway of this generalization is that it can be beneficial for an algorithm to delay dirty-oracle calls for clean-oracle calls, depending on $p$ and $r$. This contrasts the previous sections, where we can meet lower bounds by computing a dirty basis upfront.

To see this, we consider two algorithms and assume for simplicity that $\mathcal{M}_d = \mathcal{M}$. The first algorithm starts with $E$ and removes elements via binary search until it reaches an independent set. It only uses clean-oracle calls of total cost $p(n - r)\lceil \log_2(n) \rceil + p$. The second algorithm computes a dirty basis and verifies it, incurring a total cost of $n + p(n - r + 1)$, as $\mathcal{M}_d = \mathcal{M}$. Thus, for small $p$ and large $r$, the first algorithm incurs less cost than the second algorithm, which is optimal among the class of algorithms which only initially use dirty-oracle calls. Specifically, having access to rank oracles, one can compute the value of $r$ upfront using one clean-oracle call and, thus, select the better algorithm.

## 4.3 Matroid intersection

In the *matroid intersection problem*, we are given two matroids $\mathcal{M}^1 = (E, \mathcal{I}^1)$ and $\mathcal{M}^2 = (E, \mathcal{I}^2)$, and we seek a maximum set of elements $X \subseteq E$ that is independent in both matroids, i.e., $X \in \mathcal{I}^1 \cap \mathcal{I}^2$.

The textbook algorithm for finding such a maximum independent set is to iteratively increase the size of a solution one by one using an augmenting-path type algorithm until no further improvement is possible. This algorithm has a running time of $O(r^2 n)$ [17]. There are faster algorithms known for matroid intersection, which run in time $O(nr^{3/4})$ [6] and in time $O(n^{1+o(1)})$ for the special case of two partition matroids [11]. (In a *partition matroid* $\mathcal{M} = (E, \mathcal{I})$, the elements are partitioned into *classes* $C_i$ with capacities $k_i$, and a set $S \subseteq E$ is independent if and only if $|C_i \cap S| \leq k_i$ holds for each $C_i$.) Here, we focus on improving the running time of the simple textbook algorithm by (i) using dirty oracles calls in each of the augmentation steps and (ii) by computing a warm-start solution using an optimal dirty solution, i.e., a feasible solution of a certain size dependent on the error.

**Matroid intersection via augmenting paths.** Our error measure is as follows: We define $\eta_1 = \{F \in \mathcal{I}_d^1 \mid F \notin \mathcal{I}^1\}$ and $\eta_2 = \{F \in \mathcal{I}_d^2 \mid F \notin \mathcal{I}^2\}$ to be the number of different sets which are independent in the dirty matroid but not independent in the clean matroid. In order to simplify the setting, we assume here that (i) the dirty matroids are supersets of the clean matroids, i.e., $\mathcal{I}^1 \subseteq \mathcal{I}_d^1$ and $\mathcal{I}^2 \subseteq \mathcal{I}_d^2$, and (ii) that the clean matroids are partition matroids. We note that in general the intersection of two partition matroids can be reduced to finding a maximum $b$-matching.

**Proposition 4.2.** *There is an algorithm that computes an optimum solution for matroid intersection using at most* $(r + 1) \cdot (2 + (\eta_1 + \eta_2) \cdot (\lceil \log_2(n) \rceil + 2))$ *clean-oracle calls.*

**Matroid intersection via warm-starting.** We show how to exploit the dirty matroids to obtain a good starting solution that is independent in both clean matroids. The idea of warm-starting using predictions has been used for other prediction models in [10, 16, 31] for problems like weighted bipartite matching or weighted matroid intersection. These results are tailored to the weighted setting and do not directly translate to improvements in the unweighted case. As error measure we adjust the removal error for matroid intersection: Let $s_d^* = \max_{S_d \in \mathcal{I}_d^1 \cap \mathcal{I}_d^2} |S_d|$ and define $\mathcal{S}_d^* = \{S_d \in \mathcal{I}_d^1 \cap \mathcal{I}_d^2 \mid |S_d| = s_d^*\}$ to be the set of optimum solutions to the dirty matroid intersection problem. We define $\eta_r = \max_{S_d \in \mathcal{S}_d^*} \min_{S_c \in \mathcal{I}^1 \cap \mathcal{I}^2} \{|S_d \setminus S_c| : S_c \subseteq S_d\}$.

Our algorithm computes an optimal solution to the dirty matroid, then greedily removes elements until we obtain a feasible solution for the clean matroid. By observing that this reverse greedy algorithm is a 2-approximation in the number of elements to be removed, $\eta_r$, we obtain the following result.

**Proposition 4.3.** *There is an algorithm that computes a feasible solution $S_c' \in \mathcal{I}^1 \cap \mathcal{I}^2$ of size $|S_c'| \geq s_d^* - 2\eta_r$ using at most $2 + 2\eta_r \cdot (1 + \lceil \log_2(n) \rceil)$ clean-oracle calls.*

## Acknowledgments and Disclosure of Funding

We thank Kevin Schewior for initial discussions on the topic of this paper.

This research is supported by the German Research Foundation (DFG) through grant no. 517912373 and by the University of Bremen with the 2023 Award for Outstanding Doctoral Supervision. Further, the first author is funded by the Deutsche Forschungsgemeinschaft (DFG, German Research Foundation) under Germany's Excellence Strategy The Berlin Mathematics Research Center MATH+ (EXC-2046/1, project ID: 390685689).

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

# Appendix

## A    Proofs omitted from Section 2

### A.1    Proof of Lemma 2.1

**Lemma 2.1.** *The simple algorithm computes a clean basis using at most $n + 1$ clean-oracle calls. Further, if $B_d \in \mathcal{B}$, it terminates using at most $n - r + 1$ clean-oracle calls.*

*Proof.* The correctness follows easily from matroid properties. If the dirty oracle is perfectly correct, i.e. $B_d \in \mathcal{B}$, then we only need to use one clean-oracle call to check whether $B_d \in \mathcal{I}$ and $n - r$ calls for checking the maximality. If $B_d \notin \mathcal{I}$, we start from the empty set and check whether to add each of the $n$ elements. Thus, the algorithm uses at most $n + 1$ clean-oracle calls in any case.    □

### A.2    Proofs omitted from Section 2.2

We give the formal description of the algorithm discussed in Section 2.2 in Algorithm 3.

---

**Algorithm 3:** Find a basis (robustified)

**Input:** dirty basis $B_d$, matroid $\mathcal{M} = (E, \mathcal{I})$, integer $k \geq 1$

**1** $B \leftarrow \emptyset$ and $S \leftarrow B_d$
**2** Fix an order for $S = \{e_1, \ldots, e_{r_d}\}$
**3 while** $S \neq \emptyset$ **do**
**4**      Re-index the elements in $S$ from 1 to $|S|$, keeping the relative order
**5**      Find the smallest index $i \in \{1, \ldots, \min\{|S|, k-1\}\}$ with $B \cup S_{\leq i} \notin \mathcal{I}$ via *linear search*
**6**      **if** *such an $i$ exists* **then**
**7**          $B \leftarrow B \cup S_{\leq i-1}$ and $S \leftarrow S \setminus S_{\leq i}$ and go to next iteration of Line 3.
**8**      **if** $|S| \leq k - 1$ *or* $B \cup S \in \mathcal{I}$ **then**
**9**          $B \leftarrow B \cup S$ and $S \leftarrow \emptyset$ and go to next iteration of Line 3
**10**     Find the smallest index $i \in \{k, \ldots, \min\{k\lceil \log_2 r_d \rceil, |S|\}\}$ with $B \cup S_{\leq i} \notin \mathcal{I}$ via *linear search*
**11**     **if** *there is no such index $i$* **then**
**12**         Find the smallest index $i \in \{k\lceil \log_2 r_d \rceil + 1, \ldots, |S|\}$ with $B \cup S_{\leq i} \notin \mathcal{I}$ via *binary search* (guaranteed to exist)
**13**     $B \leftarrow B \cup S_{\leq i-1}$ and $S \leftarrow S \setminus S_{\leq i}$
**14 for** $e \in E \setminus B_d$ **do**
**15**     **if** $B + e \in \mathcal{I}$ **then**
**16**         $B \leftarrow B + e$
**17 return** $B$

---

This algorithm is essentially the same as the *error-dependent algorithm* from Section 2.1 but uses a robust way of finding removals. The parameter $k$ measures the trade-off between linear searches and binary searches. Note that each linear search has two parts (Line 5 and Line 10). Between the two parts, we insert a check $B \cup S \in \mathcal{I}$ (Line 8), which is necessary to prevent using too many queries when there are no further removals necessary. A small detail here is that we only have to do this check if there actually are elements to check in the second part.

**Theorem 2.3.** *For every $k \in \mathbb{N}_+$, there is an algorithm that, given a dirty matroid $\mathcal{M}_d$ of rank $r_d$ with unknown $\eta_A$ and $\eta_R$, computes a basis of a matroid $\mathcal{M}$ of rank $r$ with at most $\min\{n - r + k + \eta_A + \eta_R \cdot (k+1)\lceil \log_2 r_d \rceil, (1 + \frac{1}{k})n\}$ oracle calls to $\mathcal{M}$.*

*Proof.* We analyze Algorithm 3. Observe that the algorithm finds a maximal independent subset of $B_d$ in the clean matroid and greedily tries to add all remaining elements, similar to the error-dependent algorithm. Hence, the correctness follows as in the proof of Lemma 2.2.

It remains to bound the number of clean-oracle calls. We first prove the error-dependent upper bound. Note that the algorithm removes $|R(B_d)| \leq \eta_R$ elements from $B_d$, and then adds $|A(B_d)| \leq \eta_A$ elements. For every removed element, the algorithm uses at most $k\lceil \log_2 r_d \rceil$ clean-oracle calls due to the two linear searches, $\lceil \log_2 r_d \rceil$ calls in the binary search, and one extra query for the check between the two linear searches. Once we have a solution $B \cup S$ that is not necessarily maximum but feasible, which we do not know yet, we need to use at most $k$ additional queries until we verify that this solution is indeed feasible: We have up to $k - 1$ queries of linear search in Line 5 and one query in Line 8. After removing elements, the algorithm makes $n - r_d$ queries in Line 15. Thus, the total number of clean-oracle calls is at most $|R(B_d)|((k+1)\lceil \log_2 r_d \rceil + 1) + n - r_d$. By plugging in $r_d - \eta_R + \eta_A = r$, we proof the stated upper bound of

$$n - r + k + \eta_A + \eta_R(k+1)\lceil \log_2 r_d \rceil.$$

Second, we prove the robust upper bound that is independent of $\eta_A$ and $\eta_R$. To this end, we partition $B_d = \{e_1, \ldots, e_{r_d}\}$ into segments separated by the removed elements. Each segment (except the last one, which we treat separately below) contains exactly one removed element at the end. If a linear search (Lines 5 and 10) finds the element to be removed, we call the corresponding segment *short*. Otherwise, we say it is *long*.

Let the total length of short segments be $L_{\text{short}}$. The cost of a short segment is equal to its length, plus one for the check between the two linear searches in Line 8 if its length is at least $k$. Thus, the total cost of all short segments is at most $(1 + \frac{1}{k})L_{\text{short}}$.

Let the total length of long segments be $L_{\text{long}}$. In a long segment, there are exactly $k\lceil \log_2 r_d \rceil$ queries done by the linear searches (Lines 5 and 10), one query due to Line 8 and at most $\lceil \log_2 r_d \rceil$ many queries by the binary search in Line 12. Thus, the total cost of a long segment is at most $(k+1)\lceil \log_2 r_d \rceil + 1$. Moreover, there can be at most $\frac{L_{\text{long}}}{k\lceil \log_2 r_d \rceil + 1}$ many long segments. Therefore, the total cost of all long segments is at most $(1 + \frac{1}{k})L_{\text{long}}$.

Finally, let $L_{\text{last}}$ be the length of the last segment, where no elements are removed. If this does not exist we set $L_{\text{last}} = 0$. When the algorithm considers this segment, the condition that $B \cup S \in \mathcal{I}$ in Line 8 is true because no further elements are removed afterwards. Regarding the other condition in Line 8, there are two cases. If $|S| \leq k - 1$, then we do not query $B \cup S \in \mathcal{I}$ and terminate the while-loop. Before that, we use exactly $L_{\text{last}}$ many oracle calls in Line 5. Otherwise, that is $|S| \geq k$, we use $k - 1$ oracle calls in Line 5 and one query in Line 8, which evaluates true and terminates the while-loop. Since $k \leq |S| \leq L_{\text{last}}$, we conclude that in both cases we use at most $L_{\text{last}}$ many oracle calls during the last segment.

Therefore, the total number of queries done in Lines 3-13 is at most

$$\left(1 + \frac{1}{k}\right)(L_{\text{short}} + L_{\text{long}}) + L_{\text{last}} \leq \left(1 + \frac{1}{k}\right)r_d.$$

Together with the $n - r_d$ oracle calls in Line 15, we conclude that the total number of oracle calls is at most

$$n - r_d + \left(1 + \frac{1}{k}\right)r_d \leq \left(1 + \frac{1}{k}\right)n.$$

This concludes the proof of Theorem 2.3. $\qquad\square$

## B  Lower bounds

We present lower bounds on the worst-case number of clean queries required by any deterministic algorithm. Note that all lower bounds hold even if the dirty oracle indeed follows a matroid. First, we show that, even if $\mathcal{M} = \mathcal{M}_d$, the worst-case number of clean queries is at least $n - r + 1$. The following lemma also shows that this is best-possible number of queries for instances with $\mathcal{M} = \mathcal{M}_d$ and the best-possible worst-case bound of $n$ queries cannot be achieved at the same time.

**Lemma B.1.** *For every rank $r \geq 1$, no deterministic algorithm can use less than $n - r + 1$ clean queries, even if $\mathcal{M} = \mathcal{M}_d$. Further, for any algorithm that uses exactly $n - r + 1$ clean queries whenever $\mathcal{M} = \mathcal{M}_d$, there is at least one instance where it uses at least $n + 1$ clean queries.*

*Proof.* Consider arbitrary $n$ and $r$ with $1 \leq r \leq n$. We give a clean matroid $\mathcal{M}$ such that the statement of the lemma holds for the instance defined by $\mathcal{M}$ and the dirty matroid $\mathcal{M}_d = \mathcal{M}$. Let $E = \{e_1, \ldots, e_n\}$ denote the ground set of elements. We define $\mathcal{M} = \mathcal{M}_d$ as a partition matroid with two classes of elements $C_1 = \{e_1, \ldots, e_{r-1}\}$ and $C_2 = \{e_r, \ldots, e_n\}$. Each set $S \subseteq E$ with $|S \cap C_1| \leq r-1$ and $|S \cap C_2| \leq 1$ is independent in $\mathcal{M} = \mathcal{M}_d$, and we denote the set of independent sets by $\mathcal{I}$ and the set of bases by $\mathcal{B}$. Each basis of this matroid completely contains $C_1$ and exactly one member of $C_2$. Thus, the rank of $\mathcal{M}$ is indeed $r$. We show that even if the algorithm knows that the underlying matroid is a partition matroid (but it does not know the exact classes), it needs at least $n - r + 1$ queries.

As we require algorithms to show some certificate for a clean basis (see Section 1.3), every feasible algorithm has to verify that (i) some $B \in \mathcal{B}_d = \mathcal{B}$ is independent in $\mathcal{M}$ and that (ii) $B \cup \{e\}$ is not independent in $\mathcal{M} = \mathcal{M}_d$ for every $e \in E \setminus B$.

Consider the instance defined above. To prove (i), an algorithm *has to* make the oracle call $B \in \mathcal{I}$ as calls to supersets of $B$ will return false and calls to sets $A$ with $B \not\subseteq A$ cannot prove that $B$ is independent.

Let $e$ denote the element of $C_2$ that is part of the basis $B$ selected by the algorithm. To prove (ii), the algorithm has to verify that the elements $\{e_r, \ldots, e_n\} \setminus \{e\}$ are all part of the same class as $e$ and that this class has capacity one. Otherwise, the queries by the algorithm would not reveal sufficient information to distinguish between the matroid $\mathcal{M}$ and a potential alternative matroid $\mathcal{M}'$ that has a third class with capacity one. However, as $B$ would not be a basis for $\mathcal{M}'$, the queries by the algorithm must reveal sufficient information to distinguish $\mathcal{M}$ and $\mathcal{M}'$.

To prove that an element $e'$ is part of the same class as $e$, an algorithm has to execute a query $A \in \mathcal{I}$ with $A \cap C_2 = \{e', e''\}$ for an element $e''$ that is already known to be in the same class as $e$. This is because queries $A \in \mathcal{I}$ with either $|A \cap C_2| = 1$, $|A \cap C_2| \geq 3$ or $e' \notin A$ would always give the same answer even for the potential alternative matroid $\mathcal{M}'$ in which $e'$ forms a class of capacity one on its own.

In total, this leads to at least $n - r + 1$ oracle calls. Further, consider the robustness case that the algorithm is given an instance with a dirty matroid as described above and a clean matroid whose only independent set is the empty set. By the above arguments, in order to use only $n - r + 1$ queries for the case $\mathcal{M} = \mathcal{M}_d$, the algorithm has to query some $B \in B_d$. But this query does not help the algorithm to find out that the only clean basis is the empty set, i.e., it still needs $n$ more queries to show the fact. $\qquad\square$

The next lemma gives a lower bound that justifies the linear dependency on $\eta_A$ in our algorithmic results. In fact, it prohibits improvements over the linear dependency unless the error grows quite large. But even then, we complement the result with a different lower bound that grows stronger with increasing number of errors.

**Lemma B.2.** *For every $\eta_A \geq 1$, there is no deterministic algorithm that uses less than*

$$\begin{cases} n - r + \eta_A & \text{if } 1 \leq \eta_A \leq \frac{n - r_d}{2} \\ 1 + \lceil \log_2 \binom{n - r_d}{\eta_A} \rceil & \text{if } \eta_A \geq \frac{n - r_d}{2} \end{cases}$$

*clean queries in the worst-case, even if it knows $\eta_A$ upfront.*

We prove the lemma by dividing the two different lower bounds into two lemmas, Lemma B.3 and Lemma B.4, which we prove individually.

**Lemma B.3.** *For every number of addition errors $\eta_A$ with $1 \leq \eta_A \leq \frac{n - r_d}{2}$, there is no deterministic algorithm that uses less than $n - r + \eta_A$ clean queries in the worst-case even if the algorithm knows the value of $\eta_A$ upfront and $\eta_R = 0$.*

*Proof.* For the statement of the lemma, we assume that the algorithm has full knowledge of $\eta_A$. However, we still require that the queries of the algorithm can be used as a certificate for a third party *without* knowledge of $\eta_A$ to find a provable clean basis. Nevertheless, the assumption that the algorithm knows $\eta_A$ still restricts the query answers that can be returned by an adversary.

Fix $\eta_A \in \{1, \ldots, \lfloor \frac{n - r_d}{2} \rfloor\}$. Consider a problem instance with the ground set of elements $E = \{e_1, \ldots, e_n\}$. Let $\mathcal{M}_d$ be a partition matroid with two classes of elements $C_1 = \{e_1, \ldots, e_{r_d}\}$ and

$C_2 = \{e_{r_d+1}, \ldots, e_n\}$. Each set $S \subseteq E$ with $|E \cap C_1| \leq r_d$ and $|E \cap C_2| = 0$ is independent in $\mathcal{M}_d$. Thus, $C_1$ is the only basis.

Consider an arbitrary algorithm for the instance. Depending on the algorithm's oracle calls, the adversary will create a clean partition matroid $\mathcal{M} = (E, \mathcal{I})$ with the three classes of elements $C_1 = \{e_1, \ldots, e_{r_d}\}$, $C_2$ and $C_3$. The elements of $N := \{e_{r_d+1}, \ldots, e_n\}$ will be distributed to $C_2$ and $C_3$ depending on the algorithm's actions, and the class capacities will be $r_d$ for $C_1$, 0 for $C_2$ and $\eta_A$ for $C_3$. The adversary will assign exactly $\eta_A$ elements to $C_3$, so the only basis of the clean matroid will be $C_1 \cup C_3$ and the addition error will be indeed $\eta_A$.

The adversary starts out with empty $C_2$ and $C_3$ and later on adds elements to these sets in a consistent manner. We distinguish between two types of oracle calls by the algorithm: (i) queries to sets $S \subseteq E$ with $|(S \cap N) \setminus C_3| \geq 2$ and (ii) queries to sets $S \subseteq E$ with $|(S \cap N) \setminus C_3| = 1$ for the current $C_3$, which is initially empty. All other oracle calls only contain elements of $(E \setminus N) \cup C_3$. For such queries the adversary has to return true.

For queries of type (ii), the queried set $S \subseteq E$ contains a single element $e \in (S \cap N) \setminus C_3$ for the current $C_3$. If $e$ has already been fixed as a member of $C_2$, the oracle call returns false. Otherwise, the adversary returns true and adds the single element of $(S \cap N) \setminus C_3$ to $C_3$. After $\eta_A$ elements of $N$ have been fixed as members of $C_3$, the adversary starts to return false on all type-(ii) and assigns the single element $e \in (S \cap N) \setminus C_3$ to $C_2$ instead.

Before we move on to type-(i) queries, we give some intuition on the strategy of the adversary for type-(ii) queries. The adversary is designed to reveal the elements $C_3$ that lead to addition errors as fast as possible if the algorithm executes type-(ii) queries. If the algorithm would only execute type-(ii) queries, then after $\eta_A$ such queries the algorithm would have identified $C_3$. Since the algorithm knows $\eta_A$, it then also knows that none of the remaining elements of $N \setminus C_3$ can be added to the basis $C_1 \cup C_3$ and, thus, has also identified $C_2$. However, in order to query a certificate that proves to a third party without knowledge of $\eta_A$ that the elements of $C_2$ can indeed not be added, the algorithm will still require $|C_2|$ additional queries. Therefore, revealing $C_3$ fast will lead to $|C_2| + |C_3| = n - r + \eta_A$ queries if the algorithm only executes type-(ii) queries. If the adversary would instead reveal the elements of $C_2$ first, then the algorithm that only executes type-(ii) queries could use the knowledge of $\eta_A$ to conclude after $n - r$ queries that $C_1 \cup C_3$ must be an independent set, as otherwise the number of errors would be too small. In contrast to the previous case, the algorithm can prove this efficiently with a single query to $C_1 \cup C_3$. Thus, revealing $C_2$ first would allow the algorithm to save queries.

For queries of type (i), the adversary will use a different strategy. If the adversary would also start by returning true for type-(i) queries, then every element of $N \setminus C_3$ that is part of the query would have to be added to $C_3$. Since there are at least two such elements by definition of type-(i) queries, a single query would prove membership of $C_3$ for at least two elements. This would allow to reduce the number of queries below $n - r + \eta_A$.

Instead, we define the adversary to return false as long as this is possible. To make this query result consistent with the clean matroid, the adversary has to add one of the queried elements to $C_2$. If $S \cap C_2 \neq \emptyset$ already holds for the queried set $S$ and the current set $C_2$, then the adversary can return false without adding a further element to $C_2$. Otherwise, we define the adversary to select two elements $e_1, e_2$ of $(S \cap N) \setminus (C_3 \cup C_2)$. These elements must exist by definition of type-(i) queries and as we assume $S \cap C_2 = \emptyset$. The adversary will later on add one of $e_1$ and $e_2$ to $C_2$ depending on the algorithms actions. Assume without loss of generality that $e_1$ is the element that appears first (or at the same time as $e_2$) in another later oracle call. Once $e_1$ appears in this second query, the adversary adds $e_1$ to $C_2$. If $C_2$ already contains $n - r_d - \eta_A$ elements, then the adversary has to add it to $C_3$ instead and answer the query accordingly.

To show the lower bound, we consider two cases: (1) $n - r_d - \eta_A$ elements are added to $C_2$ before $\eta_A$ elements are added to $C_3$ and (2) $\eta_A$ elements are added to $C_3$ before $n - r_d - \eta_A$ elements are added to $C_2$. If we show that the algorithm in both cases needs at least $n - r + \eta_A$ queries, the lemma follows.

**Case 1.** First, assume the algorithm queries in such a way that $n - r_d - \eta_A$ elements are assigned to $C_2$ before $\eta_A$ elements are assigned to $C_3$. Since this is the case, an element $e$ is only added to $C_2$ if there was a type-(i) query for which the adversary selected $\{e, e'\}$ with $e \neq e'$ to potentially add to $C_2$. By definition of the adversary, each such element $e$ is afterwards added to $C_2$ while appearing in

a second query after the type-(i) query. Thus, $e$ must appear in at least two queries. Note that these two queries are distinct for each $e \in C_2$ because the assignment of $e$ is fixed immediately once $e$ appears in a second query after being selected by the adversary as part of the pair $\{e, e'\}$ in the type-(i) query. In particular, for $e'$ to also be added to $C_2$, it would need to be selected by the adversary in another type-(i) query and appear in another query afterwards. Since $|C_2| = n - r_d - \eta_A$, this leads to a total number of queries of at least $2 \cdot (n - r_d - \eta_A)$. As $\eta_A \leq \frac{n-r_d}{2}$, we have $2 \cdot (n - r_d - \eta_A) \geq 2 \cdot (n - r_d) - (n - r_d) = n - r_d = n - r + \eta_A$. If there exist elements that are never selected by the adversary in response to a type-(i) query, then the adversary must add those elements to $C_3$ as $C_2$ is already full. Since the algorithm knows $\eta_A$, it potentially knows that this is the case and can prove that these elements are in $C_3$ with a single additional query to $C_1 \cup C_3$.

**Case 2.** Next, assume the algorithm queries in such a way that $\eta_A$ elements are added to $C_3$ before $n - r_d - \eta_A$ elements are added to $C_2$. Then, the algorithm must have executed $\eta_A$ type-(ii) queries to sets $S$ where the unique element of $(S \cap N) \setminus C_3$ was added to $C_3$ afterwards.

Each element $e \in C_2$ must appear in a query to a set $S$ for which each other member is either part of $C_1$ or $C_3$ (or eventually added to $C_3$). As the algorithm knows $\eta_A$, it might be able to conclude that none of the elements in $N \setminus C_3$ can be added to the independent set $C_1 \cup C_3$ *without* actually querying them. However, the queries of the algorithm must be a certificate for a third party without knowledge of $\eta_A$. Thus, despite knowing that none of these elements can be added, the algorithm still has to prove it using type-(ii) queries. This leads to a total of at least $n - r_d - \eta_A + \eta_A = n - r + \eta_A$ queries. $\qquad \square$

Since the previous lemma only holds for errors of at most $\frac{n-r_d}{2}$, we complement it with the following lower bound for large errors.

**Lemma B.4.** *For every number of removal errors $\eta_A$ with $1 \leq \eta_A$, there is no deterministic algorithm that uses less than $1 + \lceil \log_2 \binom{n-r_d}{\eta_A} \rceil$ queries in the worst-case, even if the algorithm knows the value of $\eta_A$ upfront.*

*Proof.* The idea is similar to Lemma B.6. Consider an instance with the dirty matroid $\mathcal{M}_d = (E, \mathcal{I}_d)$ for the ground set of elements $\{e_1, \dots, e_n\}$. Let $\mathcal{M}_d$ be a partition matroid with the two classes $C_1 = \{e_1, \dots, e_{r_d}\}$ and $C_2 = \{e_{r_d+1}, \dots, r_n\}$. Each set $I$ with $|I \cap C_1| \leq r_d$ and $|I \cap C_2| \leq 0$ is independent in $\mathcal{M}_d$. Thus, $C_1$ is the only dirty basis.

Fix a number of addition errors $\eta_A \geq 1$. The adversary will create a clean matroid $\mathcal{M} = (E, \mathcal{I})$ by selecting $\eta_A$ elements from $C_2$ and add to $C_1$ instead. Formally, the clean matroid will be a partition matroid with the two classes $C_1' = C_1 \cup A$ and $C_2' = C_2 \setminus A$ for some set $A \subseteq C_2$ with $|A| = \eta_A$. The capacities for the classes are $r_d + \eta_A$ and $0$, respectively.

Firstly, the algorithm has to query $C_1$, otherwise it cannot know whether there's some element in $C_1$ that should be removed (it does not know $\eta_R$). Then, the adversary has to identify the set $A$ (or equivalently $C_2 \setminus A$) in order to execute a final query to the clean basis $C_1 \cup A$. In order to do so, it has to use queries of the form $D_1 \cup D_2$ where $D_1 \subseteq C_1$ and $D_2 \subseteq C_2$. The adversary has $\binom{|C_2|}{\eta_A}$ choices to decide $A$, and each of the queries by the algorithm can rule out at most one half of these choices (the adversary can always choose to answer "yes" or "no" depending on which preserves more choices). Thus the algorithm needs at least $\lceil \log_2 \binom{|C_2|}{\eta_A} \rceil = \lceil \log_2 \binom{n-r_d}{\eta_A} \rceil$ queries. Note that this bound is weaker than that in Lemma B.3 when $\eta_A$ is small compared to $n - r_d$. $\qquad \square$

We show that there exists an improved algorithm in terms of the error dependency if $\eta_R = 0$ and $\eta_A \geq \frac{3}{4}(n - r_d) + 1$ (cf. Lemma B.5). Hence, a distinction depending on $\eta_A$ in the lower bound (as in Lemma B.2) is indeed necessary.

**Lemma B.5.** *For problem instances with $\eta_R = 0$ and $\eta_A \geq \frac{3}{4}(n-r_d)+1$, there exists an algorithm that executes strictly less than $n - r + \eta_A$ clean queries in the worst-case.*

*Proof.* Let $B$ be a basis of the given dirty matroid $\mathcal{M}_d$. Let $N = \{e_1, \dots, e_{n-r_d}\}$ denote the elements of $E \setminus B$ indexed in an arbitrary order. Note that $|N| \geq 1$ as $\eta_R = 0$ and $\eta_A \geq 1$. Consider the following algorithm:

1. If $|N| \geq 2$, continue with Step 2. Otherwise, query $B \cup \{e\} \in \mathcal{I}$ for the single element $e \in N$. Add $e$ to $B$ if this query returns true. Afterwards, terminate.

2. Pick two arbitrary distinct elements $e, e'$ of $N$ and execute the oracle call $B \cup \{e, e'\} \in \mathcal{I}$. Remove both elements from $N$.

3. If the oracle call returns true, then add $\{e, e'\}$ to $B$.

4. Otherwise, query $B \cup \{e\} \in \mathcal{I}$. If this returns true, add $e$ to $B$. Otherwise, query $B \cup \{e'\} \in \mathcal{I}$. If this returns true, add $e'$ to $B$.

5. If $N$ is empty now, then terminate. Otherwise, go to Step 1.

We show that this algorithm satisfies the asserted statement.

Assume for now that $|N|$ is even and, thus, no queries are executed in Step 1. We call one execution of Steps 1 to 5 an iteration. Let $k_i$ with $i \in \{0, 1, 2\}$ denote the number of iterations in which $i$ many elements have been added to $B$. Then, the number of queries executed by the algorithm is at most $3 \cdot (k_0 + k_1) + k_2$.

Since $(n - r_d) - \eta_A$ elements of $N$ are not part of the basis $B$ computed by the algorithm, we must have $2 \cdot k_0 + k_1 = (n - r_d) - \eta_A$ as $k_0$ iterations identify two such elements each and $k_1$ iterations identify one such element each. Similarly, we have $k_1 + 2 \cdot k_2 = \eta_A$. By combining the two equalities, we obtain $k_0 + k_1 + k_2 = \frac{n - r_d}{2}$. Hence, the number of queries executed by the algorithm is at most

$$3 \cdot (k_0 + k_1) + k_2 = \frac{n - r_d}{2} + 2k_0 + 2k_1 \leq \frac{n - r_d}{2} + 2(2k_0 + k_1)$$

$$= \frac{n - r_d}{2} + 2(n - r_d - \eta_A) \leq \frac{n - r_d}{2} + 2\left(\frac{n - r_d}{4} - 1\right)$$

$$= n - r_d - 2 = n - r + \eta_A - 2.$$

The last inequality uses our assumed lower bound on $\eta_A$.

We note that there is a straightforward example which shows that this analysis is tight, where in particular $k_0 = 0$ and the algorithm alternatingly adds two and one elements.

Finally, if $|N|$ is odd, we have one additional query in Step 1 of the algorithm. Thus, the number of queries is in this case at most $n - r + \eta_A - 1$. $\qquad \square$

Finally, we justify the logarithmic error-dependency on the removal error $\eta_R$ in our algorithmic results for small $\eta_R$.

**Lemma B.6.** *For every $\eta_R \geq 1$, there is no deterministic algorithm that uses less than $n - r + 1 + \lceil \log_2 \binom{r_d}{\eta_R} \rceil$ clean queries in the worst-case even if the algorithm knows the value of $\eta_R$ upfront. Further, if $\eta_R \leq \frac{r_d}{2^k}$ for some fixed $k \geq 0$, the bound is at least $n - r + 1 + \lceil \eta_R(\log_2 r_d - k) \rceil$.*

*Proof.* Consider an instance with the dirty matroid $\mathcal{M}_d = (E, \mathcal{I}_d)$ for the ground set of elements $\{e_1, \ldots, e_n\}$. Let $\mathcal{M}_d$ be a partition matroid with the two classes $C_1 = \{e_1, \ldots, e_{r_d}\}$ and $C_2 = \{e_{r_d+1}, \ldots, e_n\}$. Each set $I$ with $|I \cap C_1| \leq r_d$ and $|I \cap C_2| \leq 0$ is independent in $\mathcal{M}_d$. Thus, $C_1$ is the only dirty basis.

Fix a number of removal errors $\eta_R \geq 1$. The adversary will create a clean matroid $\mathcal{M} = (E, \mathcal{I})$ by selecting $\eta_R$ elements to remove from $C_1$ and add to $C_2$ instead. Formally, the clean matroid will be a partition matroid with the two classes $C_1' = C_1 \setminus R$ and $C_2' = C_2 \cup R$ for some set $R \subseteq C_1$ with $|R| = \eta_R$. The capacities for the classes are $r_d - \eta_R$ and 0, respectively.

Consider an arbitrary deterministic algorithm. For each $e \in C_2 = C_2' \setminus R$, the algorithm has to query a set $S$ with $S \cap C_2 = \{e\}$ and $S \setminus \{e\} \subseteq C_1'$, as this is the only way of verifying that the elements in $C_2$ cannot be added to the basis. Even if the algorithm knows that $\eta_A = 0$, it still has to prove this by using such queries. This leads to at least $n - r_d \geq n - r + 1$ queries.

For the elements in $R$, the algorithm has to do the same. However, it does not know which elements of $C_1$ belong to $R$ and which do not. Instead, the algorithm has to iteratively query subsets of $C_1$

until it has sufficient information to identify the set $R$. Note that queries to sets containing members of $C_1$ *and* $C_2$ always return false independent of whether the queried members of $C_1$ belong to $R$ or not, which means that such queries do not reveal any information helpful for identifying $R$.

We can represent each such algorithm for identifying $R$ as a search tree. A vertex $v$ in the search tree represents the current state of the algorithm and is labeled by the set $P_v \subseteq C_1$ of elements that still could potentially belong to $R$ given the information already obtained by the algorithm and a family of subsets $\mathcal{S}_v$ such that each $S \in \mathcal{S}_v$ is guaranteed to contain at least one member of $R$ given the algorithm's information. For the root $r$ of the search tree, we have $P_r = C_1$ and $\mathcal{S}_r = \{C_1\}$. As the algorithm executes queries to sets $S \subseteq C_1$, it gains more information about which elements could potentially belong to $R$. Thus, each vertex $v$ in the search tree has (up-to) two successors $v', v''$ with different labels $P_{v'}$ and $P_{v''}$ depending on the answer to the query executed by the algorithm while being in the state represented by $v$. The algorithm can only terminate once it reaches a state $u$ in the search tree with $|P_u| = \eta_R$, because only then it has identified the set $R = P_u$. We refer to such vertices as *leaves*. Since, depending on the query results returned by the adversary, each subset of $S \subseteq C_1$ with $|S| = \eta_R$ could be the set $R$, there are at least $\binom{|C_1|}{\eta_R} = \binom{r_d}{\eta_R}$ leaves.

The search tree representing the algorithm has a maximum out-degree of two and at least $\binom{r_d}{\eta_R}$ leaves, which implies that the depth of the search tree is at least $\lceil \log_2 \binom{r_d}{\eta_R} \rceil$. Since the adversary can select the query results in such a way that it forces the algorithm to query according to a longest root-leaf-path, this implies that the algorithm needs at least $\lceil \log_2 \binom{r_d}{\eta_R} \rceil$ queries to identify $R$.

In total, the algorithm has to execute at least $n - r + 1 + \lceil \log_2 \binom{r_d}{\eta_R} \rceil$ queries. Note that $\binom{n}{k} \geq \frac{n^k}{k^k}$ for any positive integer $n, k$ with $k \leq n$. If $\eta_R$ is small compared to $r_d$, say, $\eta_R \leq \frac{r_d}{2^k}$ for some constant $k \geq 0$, then $\log_2 \binom{r_d}{\eta_R} \geq \eta_R(\log_2 r_d - k)$. $\qquad\square$

## C    Proofs omitted from Section 3

**Lemma 3.2** (Property (ii))**.** *At the start (end) of each iteration of Line 5, it holds* $(B_d \setminus R) \cup A \in \mathcal{I}$.

*Proof.* At the beginning of the first iteration, $(B_d \setminus R) \cup A \in \mathcal{I}$ by the condition in Line 2. It suffices to show if $(B_d \setminus R) \cup A \in \mathcal{I}$ holds before iteration $\ell = j$, then it also holds before iteration $\ell = j + 1$. If in iteration $\ell = j$, the condition in Line 6 is not satisfied, then $A, R$ do not change and we are done. The remaining case is that $e_j$ is added to $A$. If then $(B_d \setminus R) \cup A \in \mathcal{I}$, Line 8 evaluates false and we are done. Otherwise, that is, $(B_d \setminus R) \cup A \notin \mathcal{I}$, we execute the binary search in Line 9 and add an element to $R$. To conclude also this case, we argue that at the end of this iteration we have that $(B_d \setminus R) \cup A \in \mathcal{I}$. Adding $e_j$ to our solution, which by induction hypothesis was independent before, creates at most one (in this case exactly one) circuit in our solution (cf. Theorem 39.35 in [32]). Since the binary search in Line 9 detects exactly this unique circuit, an element of the circuit is added to $R$, which breaks the circuit and makes $(B_d \setminus R) \cup A$ independent again. $\qquad\square$

**Lemma 3.3** (Property (i))**.** *At the end of every iteration $\ell$ of Line 5, $(B_d \setminus R) \cup A$ is $\ell$-safe.*

*Proof.* First, we introduce some notations. Let $A_k, R_k$ be the sets represented by the variables $A, R$ at the end of iteration $k$ (equivalently at the beginning of iteration $k + 1$). We prove by induction over the iterations of Line 5 that the statement holds at the end of every iteration. First, note that every set is 0-safe, thus, this also applies to our solution before the first iteration of Line 5. Let $\ell \in \{1, \ldots, n\}$. We write $B = (B_d \setminus R_{\ell-1}) \cup A_{\ell-1}$ for our preliminary solution after iteration $\ell - 1$, and $B' = (B_d \setminus R_\ell) \cup A_\ell$ for the solution after iteration $\ell$. By induction hypothesis, we can assume that $B$ is $(\ell - 1)$-safe, and we now prove that $B'$ is $\ell$-safe. To this end, we distinguish three cases concerning element $e_\ell$.

In the first case we assume that $e_\ell \notin B_d$, Then, the algorithm makes one more query (Line 6). If $B_{\leq \ell} + e_\ell \notin \mathcal{I}$, it must hold that $B$ is $\ell$-safe, and since in this case $B = B'$, we are done. Otherwise, that is, $B_{\leq \ell} + e_\ell \in \mathcal{I}$, we add $e_\ell$ to $A$ and, thus, $B_{\leq \ell} + e_\ell = B'_{\leq \ell}$. We now prove that $B'_{\leq \ell}$ is part of a maximum-weight basis, which implies that $B'$ is $\ell$-safe. Let $B^*$ be some optimal basis such that $B_{< \ell-1} \subseteq B^*$. Such a basis exists because $B$ is $(\ell - 1)$-safe by assumption. If $e_\ell \in B^*$, then also $B'_{\leq \ell} \subseteq B^*$, and we are done. Otherwise, that is, $e_\ell \notin B^*$, there must be a circuit $C$ in $B^* + e_\ell$.

Moreover, $C$ must contain some other element $e_k$ with $k > \ell$, because $B'_{\leq\ell} \in \mathcal{I}$ due to Lemma 3.2. By the ordering of the elements, $w_{e_\ell} \geq w_{e_k}$, and therefore $B^* + e_\ell - e_k$ is a maximum-weight basis. Further, it contains $B'_{\leq\ell}$. Thus, we conclude $B'_{\leq\ell} \in \mathcal{I}^*$.

For the second case, suppose that $e_\ell \in B_d \setminus R_{\ell-1}$. Thus, the condition in Line 6 evaluates false, and we have $B = B'$. Thus, $B'_{\leq\ell} = B_{\leq\ell-1} + e_\ell$, and we can prove analogously to the previous case that $B'_{\leq\ell}$ in included in a maximum-weight basis.

Finally, it remains the case where $e_\ell \in R_{\ell-1}$. In this case, $e_\ell$ is added to $R$ in some earlier iteration $\ell' < \ell$, and, thus, $e_\ell \notin B'$. Also, $B = B'$ because $e_\ell \in B_d$. We show that $B'_{\leq\ell-1} + e_\ell$ is dependent, which implies that $B'$ is $\ell$-safe. To this end, we prove for $q = \ell', \ldots, \ell - 1$ the invariant that $((B_d \setminus R_q) \cup A_q)_{\leq\ell-1} + e_\ell$ contains a circuit $C$ such that $e_\ell \in C$ and that $e_\ell$ is the element with the largest index in $C$. Note that the invariant holds at the time $e_\ell$ is added to $R$ in iteration $\ell'$ due to Lines 3 and 10. Suppose the invariant holds for $q = j - 1$, witnessed by circuit $C_1$. We now prove the invariant for $q = j$. If $C_1 \subseteq ((B_d \setminus R_j) \cup A_j)_{\leq\ell-1} + e_\ell$, we are done. Otherwise, the algorithm must have added element $e_j \in C_1$ to $R$ in iteration $j$. By Lines 2 and 9, $e_j$ is the element with largest index in a circuit $C_2$ in $(B_d \setminus R_{j-1}) \cup A_j$. As $e_\ell$ has the largest index in $C_1$ by induction hypothesis and $e_j \in C_1$, we conclude $j < \ell$. Thus, $e_\ell \notin C_2$. Since $e_j \in C_1 \cap C_2$ and $e_\ell \in C_1 \setminus C_2$, there exists a circuit $C_3 \subseteq (C_1 \cup C_2) \setminus \{e_j\}$ such that $e_\ell \in C_3$ (cf. Theorem 39.7 in [32]). In particular, $C_3 \subseteq ((B_d \setminus R_j) \cup A_j)_{\leq\ell-1} + e_\ell$ and $e_\ell$ has the largest index in $C_3$. This makes $C_3$ a witness for the invariant for $q = j$, and, thus, concludes the proof. $\qquad\square$

**Lemma 3.5.** *It holds that $|A| \leq \eta_A$ and $|R| \leq \eta_R$.*

*Proof.* Let $A^* = A(B_d)$, $R^* = R(B_d)$ be some minimum modification sets for $B_d$ and $I^* = (B_d \cup A^*) \setminus R^*$. We choose $A^*$ and $R^*$ such that $|R \triangle R^*| + |A \triangle A^*|$ is minimized. By definition, $I^* \in \mathcal{B}^*$, and we have $|R^*| \leq \eta_R$ and $|A^*| \leq \eta_A$.

Let $I$ denote the solution computed by the algorithm. If all elements have distinct weights, it is well known that a maximum-weight basis is unique. Since $I$ is $n$-safe by Corollary 3.4, no elements can be added to $I$ while maintaining independence in $\mathcal{M}^*$. Hence, $I = I^*$. Further, since $R \cap A = \emptyset$, we have $|R| = |R^*|$ and $|A| = |A^*|$, which asserts the statement in this case. The same argumentation holds if some elements have the same weight and $I = I^*$ holds.

Hence, we assume from now on $I \neq I^*$ for the sake of contradiction. Let $\kappa < n$ be the number of distinct weights. We partition $E$ according to weight into *weight classes* of elements with the same weight. We use subscript $i$ on any set of elements to refer to the subset of elements of the $i$th weight class. Since $I$ is $n$-safe by Corollary 3.4, both $I$ and $I^*$ have maximum weight. Using again the observation that $A \cap R = \emptyset$, we have that

$$|R_i| - |A_i| = |R_i^*| - |A_i^*| \text{ and } |I_i| = |I_i^*| \tag{1}$$

for every $i \in [\kappa]$.

We now either find new modification sets $\tilde{R}^*$ and $\tilde{A}^*$ with $|R \triangle \tilde{R}^*| + |A \triangle \tilde{A}^*| < |R \triangle R^*| + |A \triangle A^*|$ or show $I^* \notin \mathcal{B}^*$; a contradiction in both cases. In the former case, we additionally require that

$$|R^*| = |\tilde{R}^*|, \ |A^*| = |\tilde{A}^*| \text{ and } \tilde{I}^* \in \mathcal{B}^*. \tag{2}$$

That is, $\tilde{R}^*$ and $\tilde{A}^*$ are minimum modification sets.

Let $i$ be the weight class of the largest weight such that $I_i \neq I_i^*$. Since $|I_i| = |I_i^*|$, we have $I_i^* \setminus I_i \neq \emptyset$.

If there exists an element $e \in (I_i^* \setminus I_i) \cap A_i^*$, there exists an element $e' \in I_i \setminus I_i^*$ with $\tilde{I}^* := I^* - e + e' \in \mathcal{B}^*$ due to the basis exchange property (cf. Theorem 39.6 in [32]). If $e' \in ((B_d)_i \setminus R_i) \setminus I_i^*$, setting $\tilde{R}^* := R^* - e'$ and $\tilde{A}^* := A^* - e$ we obtain $\tilde{I}^* := (B_d \setminus \tilde{R}^*) \cup \tilde{A}^* \in \mathcal{B}^*$. However, this contradicts our assumption that $A^*$ and $R^*$ are minimum modification sets for $B_d$ since $|\tilde{R}^*| + |\tilde{A}^*| < |R^*| + |A^*|$. Thus, $e' \in A_i \setminus I_i^*$. Therefore, setting $\tilde{R}^* := R^*$ and $\tilde{A}^* := A^* - e + e'$ implies (2) and contradicts our minimality assumption on $|R \triangle R^*| + |A \triangle A^*|$ since $e' \in A \setminus A^*$.

Otherwise, it must hold that $A_i^* \subseteq A_i$, and there must exist an element $e \in (I_i^* \setminus I_i) \cap ((B_d)_i \setminus R_i^*)$. If $A_i^* = A_i$, there must exist an element $e' \in ((B_d)_i \setminus R_i) \cap R_i^*$ such that $I^* - e + e' \in \mathcal{B}^*$ by

the basis exchange property. Setting $\tilde{R}_i^* = R_i^* - e' + e$ and $\tilde{A}^* = A^*$ in this case implies (2) and contradicts again our minimality assumption on $|R \triangle R^*| + |A \triangle A^*|$. Therefore, $A_i^* \subsetneq A_i$ and, in particular, $|A_i^*| < |A_i|$. Due to (1) and (2), $|R_i^*| < |R_i|$. Observe that for all $1 \le j \le i-1$, we have by our choice of $i$ that $I_j^* = I_j$. In the prefix $P = \bigcup_{j=1}^{i-1} I_j \cup (B_d)_i$, our algorithm guarantees that the largest independent set in $P$ has size $|P| - |R_i|$. This implies $P \setminus R_i^* \notin \mathcal{I}$, which contradicts our assumption that $I^*$ is a basis. $\qquad\square$

**Lemma 3.6.** *Algorithm 1 computes a max-weight basis with at most $n - r + 1 + 2\eta_A + \eta_R \cdot \lceil \log_2(r_d) \rceil$ clean queries.*

*Proof.* The correctness follows from Corollary 3.4. It remains to bound the number of clean queries. Note that we use clean-oracle calls only in Lines 2,3,6,8 and 9.

In Lines 3 and 9, each removal incurs a binary search, which costs at most $\lceil \log_2(r_d) \rceil$ queries. Since every binary search increments the size of $R$, the total number of queries used in these lines is at most $\lceil \log_2(r_d) \rceil \cdot |R|$. In Line 2, the number of queries is equal to the number of elements added to $R$ in Line 4 plus a final one where the condition evaluates false, which we charge extra. Similarly, we charge the queries in Line 8 to the removals in Line 10 if Line 8 holds and to the added element in Line 7 otherwise. In Line 6, we have that each $e \in E \setminus B_d$ incurs a query, hence the total number is $n - r_d$.

Summarized, the total number of clean queries is at most $\lceil \log_2(r_d) \rceil \cdot |R| + |R| + 1 + |A| + n - r_d$. Using $r = r_d + \eta_A - \eta_R$ and Lemma 3.5, we conclude the proof. $\qquad\square$

**Lemma 3.7.** *Every deterministic algorithm for finding a maximum-weight basis executes strictly more than $n - r + \eta_A + \eta_R \cdot \lceil \log_2(r_d) \rceil + 1$ clean-oracle calls in the worst-case.*

*Proof.* Consider the ground set of elements $E = \{e_1, \ldots, e_n\}$ with weights $w_{e_i} = n + 1 - i$ for all $e_i \in E$. Let the dirty matroid $\mathcal{M}_d$ be a partition matroid that is defined by the two classes $C_1 = \{e_1, \ldots, e_{n-1}\}$ and $C_2 = \{e_n\}$ with capacities $n - 2$ and 1, respectively. That is, each set $I \subseteq E$ with $|I \cap C_1| \le n - 2$ and $|I \cap C_2| \le 1$ is independent in $\mathcal{M}_d$. Then, $B_d = E \setminus \{e_{n-1}\}$ is the only dirty maximum-weight basis. Furthermore, if $\mathcal{M} = \mathcal{M}_d$, then the only way of verifying that $B_d$ is indeed a maximum-weight clean basis is to query $B_d$ and $\{e_1, \ldots, e_{n-1}\}$. The first query is necessary to prove independence and the second query is necessary to prove that $B_d$ is a maximum-weight basis. Note that querying $B_d$ and $E$ would also suffice to prove that $B_d$ is a basis, but it would not exclude the possibility that $\{e_1, \ldots, e_{n-1}\}$ is also a basis with strictly more weight. Thus, the queries $B_d$ and $E$ are not a certificate for proving that $B_d$ is a maximum-weight basis.

Consider an arbitrary deterministic algorithm. We prove the statement by giving an adversary that, depending on the algorithms clean-oracle calls, creates a clean matroid $\mathcal{M}$ that forces the algorithm to execute strictly more clean-oracle calls than the bound of the lemma.

First, we can observe that if $\mathcal{M} = \mathcal{M}_d$, we have $n - r + \eta_A + \lceil \log_2(r_d) \rceil \cdot \eta_R + 1 = 2$. This means that if the algorithm starts by querying anything but $B_d$ or $P := \{e_1, \ldots, e_{n-1}\}$, the adversary can just use $\mathcal{M}_d$ as the clean matroid. By the argumentation above, the algorithm then has to also query $B_d$ and $P$, leading to $3 > 2$ queries. Thus, it only remains to consider algorithms that start by querying either $B_d$ or $P$.

**Case 1:** The algorithm queries $P$ first. In this case, the adversary will return false. Note that this answer is consistent with the dirty matroid $\mathcal{M}_d$. This implies that the algorithm has to query $B_d$ next, as the adversary can otherwise select $\mathcal{M} = \mathcal{M}_d$, which again forces the algorithm to also query $B_d$. Thus, the algorithm would execute $3 > 2$ queries.

Consider the case where the second query of the algorithm goes to $B_d$. We define the adversary to return that $B_d$ is not independent.

Instead, the adversary will select a clean partition matroid that is defined by the three classes $C_1' = \{e_1, \ldots, e_{n-2}\} \setminus \{\bar{e}\}$ for an element $\bar{e} \in \{e_1, \ldots, e_{n-2}\}$ that is selected by the adversary in response to the further clean-oracle calls by the algorithm, $C_2' = \{\bar{e}, e_{n-1}\}$ and $C_3' = \{e_n\}$. The capacities will be $n - 3$, 0 and 1, respectively. This implies $\eta_R = 1$ as the element $\bar{e}$ induces a removal error, and $\eta_A = 0$ as $e_{n-1}$ is still not part of any maximum-weight basis, and $n - r = 2$. The bound of the lemma then becomes $n - r + \eta_A + \lceil \log_2(r_d) \rceil \cdot \eta_R + 1 = 3 + \lceil \log_2(r_d) \rceil$.

Following the proof of Lemma B.6, the adversary can force the algorithm to use $\lceil \log_2 \binom{r_d-1}{1} \rceil = \lceil \log_2 (r_d - 1) \rceil$ oracle calls to find the element $\bar{e}$. If we pick $r_d$ as a sufficiently large power of two, we get $\lceil \log_2 (r_d - 1) \rceil = \lceil \log_2 r_d \rceil$.

Note that queries containing $e_{n-1}$ do not contribute to finding $\bar{e}$ as they always return false. This implies that the algorithm needs an additional query containing $e_{n-1}$ to prove that $(B_d \setminus \{\bar{e}\}) \cup \{e_{n-1}\}$ is not independent in the clean matroid. Combined with the two queries to $B_d$ and $P$ and the $\lceil \log_2 r_d \rceil$ queries for finding $\bar{e}$, this already leads to $3 + \lceil \log_2(r_d) \rceil$ queries. Any additional query would imply that the algorithm executes strictly more queries than the bound of the lemma.

The algorithm however needs one additional query to verify that $B_d \setminus \{\bar{e}\}$ is independent. Note that the query $B_d \setminus \{\bar{e}\}$ could in principle be executed during the search for $\bar{e}$. However, the adversary would only answer true to a query of form $B_d \setminus \{e\}$ for any $e \in B_d$ if $e$ is the only remaining choice for $\bar{e}$ since returning true to that query would reveal $e = \bar{e}$, which the adversary never does. However, if $e$ is the only remaining choice for $\bar{e}$, then algorithm already knows $\bar{e}$, which means that the query is not counted within the $\lceil \log_2 r_d - 1 \rceil = \lceil \log_2 r_d \rceil$ for finding $\bar{e}$. In total, this leads to $4 + \lceil \log_2 r_d \rceil > 3 + \lceil \log_2 r_d \rceil$ queries.

**Case 2:** The algorithm queries $B_d$ first. Then, the adversary will return true, which is consistent with the dirty matroid. In order to avoid three queries in case that the adversary chooses $\mathcal{M} = \mathcal{M}_d$, the algorithm has to query $P$ next. The adversary will answer that $P$ is indeed independent and choose $E$ as the only basis of the clean matroid. Afterwards, the algorithm has to query also $E$ to prove that $E$ is indeed independent. This leads to a total of three queries. However, we have $n - r = 0, \eta_A = 1$ and $\eta_R = 0$, which implies $n - r + \eta_A + \lceil \log_2(r_d) \rceil \cdot \eta_R + 1 = 2 < 3$. $\qquad \square$

**Theorem 3.8.** *For any $k \in \mathbb{N}_+$, there is an algorithm that, given a dirty matroid $\mathcal{M}_d$ of rank $r_d$ with unknown $\eta_A$ and $\eta_R$, computes a maximum-weight basis of a matroid $\mathcal{M}$ of rank $r$ with at most $\min\{n - r + k + \eta_A \cdot (k + 1) + \eta_R \cdot (k + 1)\lceil \log_2 r_d \rceil, (1 + \frac{1}{k})n\}$ oracle calls to $\mathcal{M}$.*

*Proof.* To prove this theorem, we consider Algorithm 2. It is not hard to see that the correctness of Algorithm 2 follows from the correctness of Algorithm 1. We now prove the stated bound on the number of clean-oracle calls.

The algorithm uses clean-oracle calls only in Lines 5, 8, 9, and 11. We separately prove the error-dependent bound and the robustness bound.

**Proof of the error-dependent bound.** We start by bounding the number of queries in Lines 5, 9, and 11:

- In Line 5, each element $e \in E \setminus B_d$ incurs a query and the total number of such queries is $n - r_d$.

- In Line 11, the binary search is incurred by elements in $R$. Thus, the total number of such queries is at most $|R|\lceil \log_2 r_d \rceil$.

- To bound the number of queries in Line 9, we observe that whenever $q$ reaches $k - 1$, we execute one such query, unless $\ell = d_{\max}$, because then we know that $((B_d \setminus R) \cup A)_{\leq \ell} = (B_d \setminus R) \cup A$ and the independence follows because the condition in Line 10 evaluated false. Therefore, it suffices to bound how often $q$ reaches $k - 1$. Note that $q$ can be decreased to $0$ only in Lines 8, 9, and 12. Lines 8 and 12 correspond to an element in $R$ and Line 9 sets the variable LS to false which must have been set to true by Line 2 or Line 5 (in which case we added an element to $A$). Hence, we conclude that the number of times $q$ reaches $k - 1$, which upper bounds the number of queries executed in Line 9, is at most $|R| + |A| + 1$.

In total, the number of clean queries in Lines 5, 9, and 11 is at most

$$n - r_d + |R|\lceil \log_2 r_d \rceil + |R| + |A| + 1. \tag{3}$$

It remains to bound the number of queries executed in Line 8. To this end, we first show $q \leq k - 2$ when the algorithm terminates. This follows from the fact that at the end of the algorithm we have a feasible solution by Corollary 3.4. Assume for contradiction that $q \geq k - 1$ at the end of the algorithm. Then at some previous iteration, we entered Line 9. Consider the iteration with largest

index in which we entered Line 9. Note that the if-statement in Line 9 has to be false as otherwise $q$ was set to 0 again. But, in particular, this implies that the current solution $(B_d \setminus R) \cup A$ in this iteration is not independent and hence the algorithm must enter Line 8 or Line 11 at some later iteration in order to output an independent solution. This implies that $q$ is set to 0 again, a contradiction.

Observe that $q$ is increased by 1 before each query executed in Line 8 and the number of times $q$ is increased is at most the total decrease of $q$ plus $k - 2$ (the value of $q$ at the end of the algorithm). As shown before, $q$ can only be decreased by $k - 1$ in Line 9 which happens at most $|A| + 1$ times, or decreased by at most $k \lceil \log_2 r_d \rceil$ in Lines 8 or 12, which happens $|R|$ times. To be more precise, Line 9 is executed at most $|A|$ (rather than $|A| + 1$) times if $q > 0$ at the end of algorithm. This follows from the fact that $q > 0$ implies LS = true at the end of the algorithm, which means Line 9 is not executed after the last execution of Line 5.

To finally bound the number of queries in Line 8 we distinguish whether $q = 0$ or $q > 0$ holds at the end of the algorithm. If $q > 0$ at the end of the algorithm, then the number of queries executed in Line 8 is at most $|A| \cdot (k - 1) + |R| \cdot k \lceil \log_2 r_d \rceil + k - 2$, where the additive $k - 2$ are caused by the final at most $k - 2$ increases of $q$ at the end of the algorithm after the last reset of $q$. If $q = 0$, then there are no final queries after the last reset of $q$. However, as argued above, Line 9 is potentially executed $(|A| + 1)$ times instead of only $|A|$ times. Thus, the number of queries in Line 8 in that case is at most $(|A| + 1) \cdot (k - 1) + |R| \cdot k \lceil \log_2 r_d \rceil$. In both cases, the number of queries in Line 8 is at most

$$|A| \cdot (k - 1) + |R| \cdot k \lceil \log_2 r_d \rceil + k - 1. \tag{4}$$

We can conclude the target bound on the number of clean queries by summing up Equation (3) and Equation (4), plugging in $r_d + |A| - |R| = r$ and using Lemma 3.5, which also holds for Algorithm 2:

$$n - r_d + |R| \lceil \log_2 r_d \rceil + |R| + |A| + 1 + |A| \cdot (k - 1) + |R| \cdot k \lceil \log_2 r_d \rceil + k - 1$$
$$= n - r_d + |R| + |A| \cdot k + |R| \cdot (k + 1) \lceil \log_2 r_d \rceil + k$$
$$= n - (r - |A| + |R|) + |R| + |A| \cdot k + |R| \cdot (k + 1) \lceil \log_2 r_d \rceil + k$$
$$= n - r_d + |A| \cdot (k + 1) + |R| \cdot (k + 1) \lceil \log_2 r_d \rceil + k$$
$$\leq n - r_d + \eta_A \cdot (k + 1) + \eta_R \cdot (k + 1) \lceil \log_2 r_d \rceil + k.$$

**Proof of the robustness bound.** Let $Q_A$ denote the set of queries of Line 5 that trigger the execution of $A \leftarrow A + e_l$, i.e., $Q_A$ contains the queries that detect an addition error. Let $Q_N$ denote the queries of Line 5 that do not trigger the execution of $A \leftarrow A + e_l$ and let $Q_R$ denote the set of all remaining queries.

First, observe that each query in $Q_N$ is incurred by a distinct element of $E \setminus (B_d \cup A)$. Thus, we have $|Q_N| = n - r_d - |A|$.

Next, we continue by bounding $|Q_A|$ and $|Q_R|$. To this end, we partition $Q_R$ into *segments* $T_i$. A segment $T_i$ contains the queries of $Q_R$ that occur after the $i$'th reset of variable $q$ but before reset $i + 1$. We use *reset* to refer to an execution of Line 2, 8, 9 or 12 that sets the variable $q$ to 0. Since we count the execution of Line 2 as a reset, segment $T_1$ contains the queries of $Q_R$ that take place between the execution of Line 2 and the first execution of Line 8, 9 or 12 (if such an execution exists).

For a segment $T_i$, let $q_i$ denote the current value of variable $q$ at the final query of the segment. We distinguish between *long*, *short* and *tiny* segments:

- A segment $T_i$ is long if it contains queries executed in Line 11.

- A segment $T_i$ is tiny if it satisfies $q_i \leq k - 1$.

- A segment is $T_i$ is short if it is neither long nor tiny.

First, consider the long segments. Let $I_{\text{long}}$ denote the index set of the long segments and let $L_{\text{long}} = \sum_{i \in I_{\text{long}}} q_i$. Each long segment $T_i$ contains $k \lceil \log_2 r_d \rceil$ queries of Line 8 since $q_i$ must increase to at least this value in order to trigger queries in Line 11 and $q$ is reset afterwards. Additionally, the segment must contain a query of Line 9 as $q_i$ increases to a value larger than $k - 1$. Finally, $T_i$ contains up-to $\lceil \log_2 r_d \rceil$ queries of Line 11. Thus, a long segment $T_i$ contains at most $(k + 1) \lceil \log_2 r_d \rceil + 1$ queries. The number of long segments is at most $\frac{L_{\text{long}}}{k \lceil \log_2 r_d \rceil}$ as each long segment $T_i$

has $q_i = k\lceil \log_2 r_d \rceil$. By using that the total number of long segments is also at most $|R|$ (since each execution of Line 11 finds a distinct removal error), we get that the total number of queries over all long segments is

$$\sum_{i \in I_{\text{long}}} |T_i| \leq \frac{L_{\text{long}}}{k\lceil \log_2 r_d \rceil}((k+1)\lceil \log_2 r_d \rceil + 1) = \left(1 + \frac{1}{k}\right) L_{\text{long}} + |I_{\text{long}}| \leq \left(1 + \frac{1}{k}\right) L_{\text{long}} + |R_{\text{long}}|,$$

where $R_{\text{long}} \subseteq R$ denotes the set of removal errors that where added to $R$ in Line 12.

Next, consider the short segments. As before let $I_{\text{short}}$ denote the index set of the short segments and let $L_{\text{short}} = \sum_{i \in I_{\text{short}}} q_i$. A short segment $T_i$ contains exactly $q_i$ queries of Line 8. Furthermore, it must contain a query of Line 9 since it is not tiny and, thus, has $q_i \geq k - 1$. This implies that $T_i$ contains $q_i + 1 = (1 + \frac{1}{q_i})q_i$ queries. As $T_i$ is not tiny, we have $(1 + \frac{1}{q_i})q_i \leq (1 + \frac{1}{k})q_i$. We can conclude that the total number of queries over all short segments is at most

$$\sum_{i \in I_{\text{short}}} |T_i| \leq \left(1 + \frac{1}{k}\right) \sum_{i \in I_{\text{short}}} q_i = \left(1 + \frac{1}{k}\right) \cdot L_{\text{short}}.$$

Finally, consider the tiny segments. As before let $I_{\text{tiny}}$ denote the index set of the tiny segments and let $L_{\text{tiny}} = \sum_{i \in I_{\text{tiny}}} q_i$. Observe that each tiny segment $T_i$ either ends with the reset in Line 14 or it ends even before the reset because the algorithm terminates before $q_i$ reached value $k - 1$. The latter case can happen at most once in the final segment. Since each reset in Line 9 sets the LS flag to false and the flag is only set to $\text{true}$ again in Line 5 when an addition error is detected, the number of tiny segments is at most $1 + |A|$.

Each tiny segment contains at most $q_i + 1$ queries (up-to $q_i$ in Line 8 and up-to one in Line 9). Let $t$ denote the index of the final segment and let $\bar{L}_{tiny} = \sum_{i \in I_{\text{tiny}} \setminus \{t\}} q_i$. Without the final segment, we get

$$\sum_{i \in I_{\text{tiny}} \setminus \{t\}} |T_i| = \sum_{i \in I_{\text{tiny}} \setminus \{t\}} (q_i + 1) = \bar{L}_{\text{tiny}} + |I_{\text{tiny}}| - 1 = \bar{L}_{\text{tiny}} + |A|.$$

We can combine the bound for the queries of these tiny segments with a bound for $|Q_A|$. Since each query of $Q_A$ detects an addition error in Line 5, we have $|Q_A| = |A|$. Furthermore, we have $\bar{L}_{\text{tiny}} + |A| = k \cdot |A|$ as there are $|A|$ tiny segments that are not the final one and each such segment $T_i$ has $q_i = k - 1$. Putting it together we achieve the following combined bound:

$$\sum_{i \in I_{\text{tiny}} \setminus \{t\}} |T_i| + |Q_A| = \bar{L}_{\text{tiny}} + |A| + |A| \leq \left(1 + \frac{1}{k}\right) \cdot (\bar{L}_{\text{tiny}} + |A|).$$

It remains to consider the final segment $T_t$. If this segment does not query in Line 9, then we have $|T_t| = q_t$. In this case, we can combine all previous bounds and use that $L_{\text{long}} + L_{\text{short}} + \bar{L}_{\text{tiny}} + q_t + |A| \leq r_d + |A| - |R_{\text{long}}|$ to conclude that the total number of queries is at most:

$$|Q_N| + |Q_R| + |Q_A| = |Q_N| + \sum_{i \in I_{\text{long}}} |T_i| + \sum_{i \in I_{\text{short}}} |T_i| + \sum_{i \in I_{\text{tiny}} \setminus \{t\}} |T_i| + |Q_A| + |T_t|$$

$$\leq n - r_d - |A| + \left(1 + \frac{1}{k}\right) \cdot (\bar{L}_{\text{tiny}} + |A| + L_{\text{short}} + L_{\text{long}}) + |R_{\text{long}}| + q_t$$

$$\leq n - r_d - |A| + \left(1 + \frac{1}{k}\right) \cdot (r_d + |A| - |R_{\text{long}}|) + |R_{\text{long}}|$$

$$\leq \left(1 + \frac{1}{k}\right)(n - r_d - |A| + r_d + |A| - |R_{\text{long}}| + |R_{\text{long}}|) = \left(1 + \frac{1}{k}\right) n.$$

Note that the last inequality holds because $|A| \leq n - r_d$.

If $T_t$ executes a query in Line 9, then we have $|T_t| = q_t + 1$. On the other hand, the query in Line 9 implies $\ell < d_{\max}$ for the current index $\ell$ when the query is executed and for $d_{\max} = \max_{e_i \in B_d} i$. This means that the variable $q$ is increased at most $r_d - |R_{\text{long}}| - 1$ times and, thus, $L_{\text{long}} + L_{\text{short}} +$

$\bar{L}_{\text{tiny}} + q_t + |A| \leq r_d + |A| - |R_{\text{long}}| - 1$. Plugging these inequalities into the calculations above yields the same result:

$$
\begin{aligned}
|Q_N| + |Q_R| + |Q_A| &= |Q_N| + \sum_{i \in I_{\text{long}}} |T_i| + \sum_{i \in I_{\text{short}}} |T_i| + \sum_{i \in I_{\text{tiny}} \setminus \{t\}} |T_i| + |Q_A| + |T_t| \\
&\leq n - r_d - |A| + \left(1 + \frac{1}{k}\right) \left(\bar{L}_{\text{tiny}} + |A| + L_{\text{short}} + L_{\text{long}}\right) + |R_{\text{long}}| + q_t + 1 \\
&\leq n - r_d - |A| + \left(1 + \frac{1}{k}\right) \left(r_d + |A| - |R_{\text{long}}| - 1\right) + |R_{\text{long}}| + 1 \\
&\leq \left(1 + \frac{1}{k}\right) \left(n - r_d - |A| + r_d + |A| - |R_{\text{long}}| - 1 + |R_{\text{long}}| + 1\right) \\
&= \left(1 + \frac{1}{k}\right) n.
\end{aligned}
$$

This concludes the proof of Theorem 3.8. $\qquad\square$

## D  Proofs omitted from Section 4

### D.1  Discussion omitted from Section 4.1

Another commonly used type of matroid oracle is the rank oracle: given any $S \subseteq E$, a rank oracle returns the cardinality of a maximum independent set contained in $S$, denoted by $r(S)$. We show in this section that a rank oracle can be much more powerful than an independence oracle for our problem. First note that since $r(S) = |S|$ if and only if $S \in \mathcal{I}$, our previous results for independence oracles immediately translate. Moreover, we can even reduce the number of oracle calls using a rank oracle. We briefly discuss key ideas for the unweighted case below. It would be interesting to see whether these ideas can also be used for the weighted case.

We start by adapting the simple algorithm from Section 2: Assume w.l.o.g. $B_d \neq \emptyset$. If $r(B_d) \neq |B_d|$ ($B_d \notin \mathcal{I}$), we run the greedy algorithm. Otherwise, we check whether $r(E) = |B_d|$. If it holds, we can conclude $B_d \in \mathcal{B}$ and are done. (This is the main difference compared to the simple algorithm with an independence oracle.) If not, we greedily try to add each element $e \in E \setminus B_d$ using in total $n - |B_d| \leq n - 1$ queries. Thus, in every case the algorithm does at most $n + 1$ clean-oracle calls. Moreover, unlike the simple algorithm with an independence oracle, it only does 2 oracle calls if $B_d \in \mathcal{B}$. In particular, this shows that Lemma B.1 does not hold for rank oracles anymore.

Using the same idea, we can improve the number of oracle calls whenever $B_d \in \mathcal{B}$ for the error-dependent algorithm from Section 2.1 to 2. Furthermore, we can also improve its worst-case number of queries from $\Theta(n \log_2 n)$ to $n + 1$. Recall that this bad case happens when $\eta_R$ is large. However, now we can simply compute $r(B_d)$ and obtain $\eta_R = |B_d| - r(B_d)$. Depending on $\eta_R$, we then decide whether to remove elements via binary-search or immediately switch to the greedy algorithm. In particular, achieving this worst-case guarantee of $n+1$ *does not* affect the error-dependent bound on the number of oracle calls.

Finally, we can improve the dependency on $\eta_A$. Recall that in the error-dependent algorithm, we iteratively augment an independent set $B$ with $\eta_A$ many elements from $E \setminus B_d$ to a basis. But with a rank oracle, we can find these elements faster via binary search: find the first prefix $(E \setminus B_d)_{\leq i}$ which satisfies $r(B \cup (E \setminus B_d)_{\leq i}) > r(B)$, and add $(E \setminus B_d)_{=i}$ to $B$. Doing this exhaustively incurs at most $\eta_A \lceil \log_2 (n - r_d) \rceil$ clean-oracle calls. Thus, whenever $\eta_A < \frac{n - r_d}{\lceil \log_2 (n - r_d) \rceil}$, this strategy gives an improved error-dependent bound over considering elements in $E \setminus B_d$ linearly. Furthermore, this condition can be easily checked with the rank oracle. Note that this contrasts Lemma B.2.

To conclude the discussion, we obtain the following proposition.

**Proposition 4.1.** *There is an algorithm that computes a clean basis with at most* $\min \left\{ n + 1, 2 + \eta_R \cdot \lceil \log_2 r_d \rceil + \min \left\{ \eta_A \cdot \lceil \log_2 (n - r_d) \rceil, n - r_d \right\} \right\}$ *clean rank-oracle calls.*

## D.2 Proofs omitted from Section 4.3

In the matroid intersection problem we are given two matroids $\mathcal{M}^1 = (E, \mathcal{I}^1)$ and $\mathcal{M}^2 = (E, \mathcal{I}^2)$ on the same set of elements $E$ and we seek to find a maximum set of element $X \subseteq E$ such that $X \in \mathcal{I}^1 \cap \mathcal{I}^2$, i.e., it is independent in both matroids. For $i \in \{1, 2\}$ we define $r_i$ to be the rank of matroid $\mathcal{I}^i$.

### D.2.1 Matroid intersection via augmenting paths

The textbook algorithm for finding such a maximum independent set is to iteratively increase the size of a solution by one and eventually reach a point in which no improvement is possible; in that case we also get a certificate: $U \subseteq E$ with $|X| \geq r_1(U) + r_2(E \setminus U)$. Given some feasible solution $X \in \mathcal{I}^1 \cap \mathcal{I}^2$, in every iteration the algorithm executes the following steps.

1. Construct the directed bipartite exchange graph $D(X)$ for sets $X$ and $E \setminus X$, where for every $x \in X$ and $y \in E \setminus X$ there is an edge $(x, y)$ if $X - x + y \in \mathcal{I}^1$ and there is an edge $(y, x)$ if $X - x + y \in \mathcal{I}^2$. Compute sets $Y_1 = \{y \in E \setminus X \mid X + y \in \mathcal{I}^1\}$ and $Y_2 = \{y \in E \setminus X \mid X + y \in \mathcal{I}^2\}$.

2. If $Y_1 = \emptyset$ terminate with certificate $U = E$. If $Y_2 = \emptyset$ terminate with certificate $U = \emptyset$. Otherwise, compute a shortest path $P$ between any vertex of $Y_1$ and any vertex of $Y_2$.

3. If no such path exists, terminate with the certificate $U \subseteq E$ of elements for which there exists a directed path in $D(X)$ to some element of $Y_2$. Otherwise, augment $X$ along $P$, that is, $X \leftarrow X \triangle P$, and continue with the next iteration.

We call the path found in step 2 an *augmenting path*. The running time of this classic algorithm is $O(r^2 n)$ [17], where $n = |E|$ and $r = \min\{r_1, r_2\}$. Recently, there has been a lot of significant progress concerning the running times for matroid intersection, culminating in the currently best running time of $O(nr^{3/4})$ for general matroids [6] and time $O(n^{1+o(1)})$ for special cases, e.g., the intersection of two partition matroids [11].

Here, we focus on improving the running-time of the simple textbook algorithm for matroid intersection, by (i) using dirty oracles calls in each of the augmentation steps or (ii) by using warm-starting ideas, i.e., by computing a feasible solution of a certain size dependent on the error using an optimal dirty solution.

Formally, additionally to the input for matroid intersection we are also given two dirty matroids $\mathcal{M}_d^1 = (E, \mathcal{I}_d^1)$ and $\mathcal{M}_d^2 = (E, \mathcal{I}_d^2)$. Our goal is to improve the classic textbook algorithm from above by using dirty-oracle calls. We attempt the following approach: Given a feasible solution $X$ (for the clean oracles), we do one iteration of the above algorithm using the dirty oracles. If there is an augmenting path, which is also an augmenting path for the clean matroids, we augment our solution and go to the next iteration. Otherwise, we found an augmenting path using dirty oracles, which is not an augmenting path for the clean matroids. In this case we do a binary search to find the error, update our dirty matroids and start the algorithm again. Finally, if there is no augmenting path in the dirty matroid, we need to augment the solution using clean-oracle calls, as we do not benefit from using the dirty oracle anymore. To avoid this situation, throughout this section we assume that the dirty matroids are supersets of the clean matroids, i.e., $\mathcal{I}^1 \subseteq \mathcal{I}_d^1$ and $\mathcal{I}^2 \subseteq \mathcal{I}_d^2$.

Further, in step 2 of the augmenting path algorithm it is crucial that the path from $Y_1$ to $Y_2$ is a path without chords[1], as otherwise the computed solution may not be feasible. Note that in step 2 we compute a shortest path, which is always a path without chords. Therefore, if we use the dirty oracles to compute a shortest path w.r.t. the dirty oracles, we need to verify in each step that the computed path has no chords for the clean matroids as well. To avoid such a situation, we restrict to the case that the clean matroids are partition matroids, as for those matroids we do not need to find an augmenting path without chords, but just *any* augmenting path. We note that in general the intersection of two partition matroids can be reduced to finding a maximum $b$-matching.

Therefore, from now on we assume that the clean matroids are partition matroids and that the dirty matroids are supersets of the clean matroids. Our algorithm works as follows. We assume that we

---

[1] A chord $e$ of some $s$-$t$ path $P$ is an edge such that $P + e$ admits an $s$-$t$ path $P'$ that is strictly shorter than $P$.

are given a feasible solution for the clean matroid and wish to increase its size by one if possible. Additionally, we are given two lists $\mathcal{F}_1$ and $\mathcal{F}_2$ of false dirty queries for $\mathcal{M}_d^1$ and $\mathcal{M}_d^2$, respectively, i.e., two list of sets $F \subseteq E$ for which we have already queried that $F$ is not independent in $\mathcal{M}^1$ or $\mathcal{M}^2$, respectively (but it was independent in the respective dirty matroid). For each iteration, we use the following algorithm.

---

**Algorithm 4:** Augmenting path via dirty oracles

---

**Input:** A feasible solution $X \in \mathcal{I}^1 \cap \mathcal{I}^2$ and two lists of false dirty queries $\mathcal{F}_1, \mathcal{F}_2$

1 Compute augmenting path $P$ in $D_d^{\mathcal{F}}(X)$
2 **if** *there is no such path $P$* **then**
3     **return** $X$ (in this case $X$ is optimal)
4 **if** *$P$ is augmenting path for clean matroids* **then**
5     $X' \leftarrow X \Delta P, \mathcal{F}_1' \leftarrow \mathcal{F}_1, \mathcal{F}_2' \leftarrow \mathcal{F}_2$
6 **if** *$P$ is not an augmenting path for clean matroids* **then**
7     find the first edge on $P$ such that $e \notin D(X)$ via *binary search*. Add corresponding set to $\mathcal{F}_1$ or $\mathcal{F}_2$ and go to Line 1.

---

Here, $D_d^{\mathcal{F}}(X)$ is the bipartite exchange graph for the dirty matroids $\mathcal{M}_d^1$ and $\mathcal{M}_d^2$, in which we have excluded the list of false queries $\mathcal{F}$. More formally, for every $x \in X$ and $y \in E \setminus X$ there is an edge $(x, y)$ in $D_d^{\mathcal{F}}(X)$ if $X - x + y \in \mathcal{I}_1$ and $X - x + y \notin \mathcal{F}_1$ and there is an edge $(y, x)$ in $D_d^{\mathcal{F}}(X)$ if $X - x + y \in \mathcal{I}_2$ and $X - x + y \notin \mathcal{F}_2$.

Next, we analyze this augmenting path algorithm. In order to define an error measure, let $\eta_1 = \{F \in \mathcal{I}_d^1 \mid F \notin \mathcal{I}^1\}$ and $\eta_2 = \{F \in \mathcal{I}_d^2 \mid F \notin \mathcal{I}^2\}$ be the number of different sets which are independent in the dirty matroid but not independent in the clean matroid. We now show the following result, which is stronger than Proposition 4.2.

**Proposition D.1.** *Let $\mathcal{M}^1$ and $\mathcal{M}^2$ be two partition matroids and $\mathcal{M}_d^1$ and $\mathcal{M}_d^2$ be two dirty matroids such that $\mathcal{M}_d^1$ and $\mathcal{M}_d^2$ are supersets of $\mathcal{M}^1$ and $\mathcal{M}^2$, respectively. Given a feasible solution $X \in \mathcal{I}^1 \cap \mathcal{I}^2$ of value $k$, there is an algorithm that either returns a solution of value $k + 1$ or outputs that $X$ is maximum using at most $2 + (\eta_1 + \eta_2) \cdot (\lceil \log_2(n) \rceil + 2)$ clean-oracle calls.*

*Moreover, there is an algorithm that computes an optimum solution for matroid intersection using at most $(r + 1) \cdot (2 + (\eta_1 + \eta_2) \cdot (\lceil \log_2(n) \rceil + 2))$ clean-oracle calls.*

*Proof.* We first prove that the solution output by the algorithm is feasible. If there is no augmenting path in $D_d^{\mathcal{F}}(X)$, then there is no augmenting path in $D(X)$ since the dirty matroids are supersets of the clean matroids and in $\mathcal{F}_1$ and $\mathcal{F}_2$ we only save queries which are not independent in the clean matroid. Once we find an augmenting path $P$ in $D_d^{\mathcal{F}}(X)$, we always verify it using two clean-oracle calls: We query if $X \Delta P$ is independent in $\mathcal{M}_1$ and in $\mathcal{M}_2$. Hence, we only augment $X$ if the augmenting path $P$ is also an augmenting path in $D(X)$. Since both clean matroids are partition matroids, we do not need to verify that $P$ is an augmenting path without chords and hence $X \Delta P$ is an independent set in both clean matroids.

It remains to prove that we use at most $2 + (\eta_1 + \eta_2) \cdot \lceil \log_2(n) \rceil$ clean-oracle calls. As described above, after finding an augmenting path $P$ in $D_d^{\mathcal{F}}(X)$, we verify it using two clean-oracle calls: We query if $X \Delta P$ is independent in $\mathcal{M}_1$ and in $\mathcal{M}_2$. If it is independent, we are done. This check needs 2 clean-oracle calls. Otherwise, one of the queries tells us that $X \Delta P$ is not independent, i.e., one of the edges $e$ in $P$ corresponds to a resulting set $X_e$ such that $X_e$ is independent in, say $\mathcal{M}_d^1$, but not independent in $\mathcal{M}^1$. In particular, in this case Line 7 is executed.

We now show the following: Whenever Line 7 is executed, we add to $\mathcal{F}_1$ or $\mathcal{F}_2$ an additional set and we can find such a set using at most $\log_2(n)$ many clean-oracle calls. Since $|\mathcal{F}_1| + |\mathcal{F}_2| \le \eta_1 + \eta_2$ by definition, this shows the desired bound.

The exchange graph $D(X)$ is a subgraph of the exchange graph $D_d^{\mathcal{F}}(X)$ since the dirty matroids are supersets of the clean matroid and by definition of $\mathcal{F}_1$ and $\mathcal{F}_2$. Hence, if $P = e_1, e_2, ..., e_p$ is an augmenting path in $D_d^{\mathcal{F}}(X)$ but not an augmenting path in $D(X)$, there must be an edge $e_i \in P$ which is not in $D(X)$. Let $e_i$ be the first such edge. In order to find $e_i$, we simply do a binary search:

In each iteration, we have two pointers $\ell$ and $r$, satisfying $0 \le \ell \le i$ and $i \le r \le p$, for which we know that the first $\ell$ edges of $P$ are also in $D(X)$ and among the edges $e_{\ell+1}, ..., e_r$ there must be an edge which is not in $D(X)$. Then, for $P' = e_1, e_2, ..., e_{\ell + \lfloor \frac{r-\ell}{2} \rfloor}$ we simply query if $X \Delta P'$ is independent in both clean matroids. If yes, then we set $\ell = \ell + \lfloor \frac{r-\ell}{2} \rfloor$. If no, we set $r = \ell + \lfloor \frac{r-\ell}{2} \rfloor$. We repeat this until we found $e_i$. This binary search algorithm finds the first edge $e \in P$ which is not in $D(X)$ using at most $\lceil \log_2(n) \rceil$ many clean-oracle calls, where $n$ is the number of elements $|E|$. The additional $+2$ in the bound appears since whenever we compute a new candidate augmenting path $P$, we first test if it is also an augmenting path in $D(X)$ using 2 clean-oracle calls. Therefore, we obtain the bound of at most $2 + (\eta_1 + \eta_2) \cdot (\lceil \log_2(n) \rceil + 2)$ clean-oracle calls.

We obtain the final bound of $(r+1) \cdot (2 + (\eta_1 + \eta_2) \cdot (\lceil \log_2(n) \rceil + 2))$ clean-oracle calls for computing an optimum solution for matroid intersection as follows: In each iteration we increase the size of the current solution by one or prove that there is no improvement possible (and hence the solution is optimal). Therefore, there are at most $r + 1$ iterations, which proves the bound. $\qquad\square$

### D.2.2 Matroid intersection via warm-starting using a dirty solution

In this subsection we consider the task of using a solution to the dirty matroid intersection instance as a "warm-start" for the clean matroid intersection instance. In particular, we compute a maximal subset of the dirty solution using only few clean-oracle calls, which again will depend on the error of the dirty matroids. Similar approaches for warm-starting using predictions have been used in [10, 16, 31] for problems like weighted bipartite matching or weighted matroid intersection. We show here that this is also possible using dirty matroids.

Before we start with the algorithm, let us define our error measure for this subsection. Here, we are just interested in the removal error: for a given maximum solution $S_d$ for the dirty matroid intersection, we compute the shortest distance to obtain a feasible solution to the clean matroid intersection problem. Then, the error is simply the maximum value of this shortest distance among all maximum solutions $S_d$ for the dirty matroid intersection. More formally, let $s_d^* = \max_{S_d \in \mathcal{I}_d^1 \cap \mathcal{I}_d^2} |S_d|$ and define $\mathcal{S}_d^* = \{S_d \in \mathcal{I}_d^1 \cap \mathcal{I}_d^2 \mid |S_d| = s_d^*\}$ to be the set of optimum solutions to the dirty matroid intersection problem. We define $\eta_r = \max_{S_d \in \mathcal{S}_d^*} \min_{S_c \in \mathcal{I}^1 \cap \mathcal{I}^2} \{|S_d \setminus S_c| : S_c \subseteq S_d\}$.

Our algorithm works as follows. We first compute a maximum solution $S_d \in \mathcal{I}_d^1 \cap \mathcal{I}_d^2$ using some algorithm for matroid intersection. Then, via binary search we greedily remove elements to first obtain a solution which is in $\mathcal{I}^1$, and then do the same to obtain a solution which is also in $\mathcal{I}^2$. We will then show that this solution satisfies $|S_c'| \ge |S_d| - 2\eta_r$ and that we use at most $2 + 2\eta_r \cdot (1 + \lceil \log_2(n) \rceil)$ many clean-oracle calls to compute $S_c'$.

---

**Algorithm 5:** Obtaining a warm-start solution

**Input:** Two dirty and clean matroids.
1 Compute maximum-cardinality solution $S_d \in \mathcal{I}_d^1 \cap \mathcal{I}_d^2$
2 **for** $i = 1, 2$ **do**
3    **if** $S_d \in \mathcal{I}^i$ **then**
4       $\lfloor$ go to Line 2 with $i = 2$ or return $S_c' = S_d$ if $i = 2$
5    **else**
6       sort elements in $S_d$ in arbitrary order $e_1, e_2, ..., e_{|S_d|}$
7       find the first element $e_j \in S_d$ such that $\{e_1, ..., e_{j-1}\} \in \mathcal{I}^i$ and $\{e_1, ..., e_j\} \notin \mathcal{I}^i$ via *binary search*. Set $S_d = S_d - e_j$ and go to Line 3

---

**Proposition 4.3.** *There is an algorithm that computes a feasible solution $S_c' \in \mathcal{I}^1 \cap \mathcal{I}^2$ of size $|S_c'| \ge s_d^* - 2\eta_r$ using at most $2 + 2\eta_r \cdot (1 + \lceil \log_2(n) \rceil)$ clean-oracle calls.*

*Proof.* Feasibility is clear since we check in Line 3 for both matroids whether the solution is indeed feasible. We first prove that we remove at most $2\eta_r$ many elements from $S_d$ to obtain a feasible solution $S_c' \in \mathcal{I}^1 \cap \mathcal{I}^2$. By the definition of the error $\eta_r$, there is some set $S_c \in \mathcal{I}^1 \cap \mathcal{I}^2$ with $S_c \subseteq S_d$ of size $|S_d| - \eta_r$. Hence, there is some set $R_\eta^1$ such that $S_d \setminus R_\eta^1 \in \mathcal{I}^1$ and some set $R_\eta^2$ such that $S_d \setminus R_\eta^2 \in \mathcal{I}^2$, where $|R_\eta^1| \le \eta_r$ and $|R_\eta^2| \le \eta_r$. Therefore, $S_c' := S_d \setminus (R_\eta^1 \cup R_\eta^2) \in \mathcal{I}^1 \cap \mathcal{I}^2$

and $|S'_c| \geq |S_d| - 2\eta_r$. Note that since $\mathcal{M}^1$ and $\mathcal{M}^2$ are matroids, the set $R^1_\eta$ can be found by greedily removing elements from $S_d$ such that $S_d \setminus R^1_\eta \in \mathcal{I}^1$ and, afterwards, the set $R^2_\eta$ can be found by greedily removing elements from $S_d \setminus R^1_\eta$ such that $(S_d \setminus R^1_\eta) \setminus R^2_\eta \in \mathcal{I}^2$. Since Algorithm 5 computes such a set, we conclude that Line 5 is executed at most $2\eta_r$ times.

Next, we show that whenever Line 5 is executed, we use at most $\lceil \log_2(n) \rceil$ many clean-oracle calls. We fix some $i \in \{1, 2\}$. Let $e_1, e_2, ..., e_{|S_d|}$ be any order of the elements. To find the first element $e_j \in S_d$ such that $\{e_1, ..., e_{j-1}\} \in \mathcal{I}^i$ and $\{e_1, ..., e_j\} \notin \mathcal{I}^i$, the algorithm performs a binary search. By folklore results, we use at most $\lceil \log_2(n) \rceil$ many clean-oracle calls to do so. Furthermore, whenever Line 5 is executed, we have previously executed a clean query in Line 3. Finally, we additionally need 2 clean-oracle calls, as Line 5 is executed for both matroids even if there is no error at all. Therefore, we use at most $2 + 2\eta_r \cdot (1 + \lceil \log_2(n) \rceil)$ many clean-oracle calls. $\qquad\square$

