# OpenReview forum: "Accelerating Matroid Optimization through Fast Imprecise Oracles"
_NeurIPS.cc/2024/Conference — NeurIPS 2024 poster_

### Official Review · Reviewer_AWmg · 2024-07-12

**Soundness:** 4
**Presentation:** 4
**Contribution:** 3
**Rating:** 6
**Confidence:** 3

**Summary:**

The paper studies the problem of finding a maximum-weight basis of a matroid in a learning-augmented setting, where there are two different oracles that the algorithm can query to check whether a set is independent: An exact oracle that always gives correct answers (but may be slow), and a dirty oracle that may give incorrect answers. Formally, the answers of the dirty oracle correspond to some other "dirty" matroid (or more generally, downward closed set system) that ideally is close to the true matroid. In the setting considered, queries to the dirty oracle are free, and the goal is to minimize the number of queries to the exact oracle to find a maximum weight basis.

The quality of the dirty oracle is parametrized by two error measures, corresponding roughly to the numbers of elements that would need to be added/removed to move from a max-weight basis for the dirty matroid to a max-weight basis for the true matroid. The main result is an algorithm whose number of clean queries interpolates (as a function of the error parameters) between n-r+k (where n is the number of items in the ground set and r the size of any basis, and $k\ge 1$ is an algorithm parameter that can be chosen) and $n(1+1/k)$. One should note that without dirty queries, the problem requires n exact queries. The dependence on the error in the interpolation is shown to be asymptotically optimal.

If the algorithm has access to rank queries rather than just independence queries (rank queries return the rank of a set, i.e., the size of the maximal independent subset) the guarantees can be improved to a quantity interpolating beetween 2 and n+1 exact queries depending on the input.

An extension of the main results includes an application to the matrix intersection problem.

**Strengths:**

- Essentially tight bounds.
- Non-trivial algorithms. They are not overly complicated, but getting everything right requires some care.
- Clear presentation. The simpler algorithm for the unweighted setting is helpful before diving into the unweighted setting.

**Weaknesses:**

- In order for the improvement of the algorithm for independence queries to be a super-constant factor requires n-r is sub-linear in n. It is not clear how realistic this is. The authors hint at graphic networks in sparse graphs having high rank -- does this mean $r=\Omega(n)$ or even $r = n - o(n)$? The result in the case of rank queries avoids this issue though.
- Significant work is required for the robustness result, i.e., to ensure the number of clean queries is simultaneously bounded by a function of the error parameters and by a quantity slightly larger than n. However, the trivial approach of running two algorithms in parallel would achieve a similar result, losing only a factor of 2, so the improvement over this trivial approach is relatively small.
- The extension to matroid intersection seems relatively weak (strong assumptions and an error measure that can be exponentially large in n in the first setting). Since this is only an extension rather than a main result, this does not affect the main contribution though.

**Questions:**

Please elaborate on the size of n-r in typical settings, in relation to the first possible weakness above.

**Limitations:**

Yes

---

> ### Author Rebuttal · Authors · 2024-08-07
>
> ### Dependence on n-r
>
> Indeed, our results on independence oracles cannot avoid the dependence on n-r, though this can be mitigated by using a stronger oracle type, such as a rank oracle.
> Nevertheless, improvements over the greedy guarantee are possible, e.g. for graphic matroids of sparse graphs.
> In the graphic matroid of a connected graph $G=(V,E)$, we have $n = |E| \ge |V|-1$ and $r = |V|-1$. If the graph is sparse and satisfies $|E| = |V|-1 + x$ for an $x \in o(|V|)$, then $n-r$ is asymptotically smaller than the greedy guarantee of $n$. Likely more realistic is a value $x \le c \cdot |V|$ for some constant $c$. While this does not give an asymptotic improvement over the greedy algorithm, saving a constant factor can be significant in practice.
>
>
> ### Our Robustness Approach vs Standard Approach
>
> It is correct that the standard approach in learning-augmented algorithms, which involves running two algorithms alternatingly (in parallel), guarantees a simple robustness by losing a factor of 2 in robustness _and_ consistency.
> Our more complex algorithm can achieve a robustness of 2n and at the same time the optimal consistency of n-r+1 (when k=1).
> Moreover, we can achieve a better-than-2 robustness while only losing an additive constant in the consistency, which is not possible with the standard approach.

---

> > ### Comment · Reviewer_AWmg · 2024-08-12
> >
> > Thank you for your response. I'm intending to keep my score.

---

### Official Review · Reviewer_rC5n · 2024-07-13

**Soundness:** 3
**Presentation:** 1
**Contribution:** 3
**Rating:** 5
**Confidence:** 1

**Summary:**

This paper aims to solve fundamental matroid optimization problems, specifically, computing a maximum-weight basis of a matroid, a complex combinatorial optimization problem, To this end, the author proposes a two-oracle model, which uses fast but dirty oracle to reduce the time to call clean oracles. Then, the author proposes the algorithm to compute the maximum-weight basis and gives the upper bound of the oracles calls needed. Finally, the author also discusses advanced settings like different kinds of oracles and other matroid optimization problem.

**Strengths:**

1. The theoretical technique of this paper seems strong, the paper gives a theorem and detailed proof of the bound of the cost to call oracles.

2. The two-model design is superior to traditional methods which require n+1 clean calls, this seems to be a huge increase.

3. The extended result in the Section 4 is interesting, it helps to show that the algorithm is generalizable and can be suitable for different settings.

**Weaknesses:**

1. The organization of this paper is very confusing, for example, the author includes a large part of the preliminary in its introduction rather than an independent section. Moreover, the author includes many extended results and propositions in its final section 4, but doesn't include a conclusion.

2. The paper studies combinatorial optimization problems which can be hard to comprehend for readers with not much background knowledge, while the paper is not self-contained enough. The preliminary in Section 1.3 alone is not enough to grasp an intuitive concept of that.

3. This paper doesn't include any empirical experiment.

**Questions:**

Perhaps it will be beneficial to include more figures as illustrations? For example, you can give an example of the algorithm walkthrough as a toy example. Also, it can also be helpful for the readers to understand the new proposed concept like k-safe and modification example.

**Limitations:**

I think the author has stated the limitations as he/she clearly stated the necessary assumptions in the respective theorem.

---

> ### Author Rebuttal · Authors · 2024-08-07
>
> ### Organization of the Paper and Figures
>
> We regret that space constraints prevent us from providing a more gentle introduction to the well-established field of matroid optimization. In a full version, we will include more figures and examples for our algorithms to better illustrate our results.
>
> ### Empirical Verification
>
> We are happy to add experiments in a full version. Our work provides theoretical results with proofs that hold true without experimental verification. It has become increasingly common in the field of learning-augmented algorithm design to omit experimental verification for purely theoretical results, even when publishing at AI venues.

---

### Official Review · Reviewer_KsE6 · 2024-07-13

**Soundness:** 3
**Presentation:** 3
**Contribution:** 3
**Rating:** 7
**Confidence:** 3

**Summary:**

The paper mainly studies the problem of finding a maximum weight basis in a matroid $\mathcal{M}=(E,\mathcal{I})$ using two types of independence oracles "clean" and "dirty". The clean oracle determines whether a set $S \subseteq E$ is an independent set in $\mathcal{M}$, and the dirty oracle determines independence according to another matroid $\mathcal{M}_d=(E,\mathcal{I}_d)$. The dirty oracle is free but might be imprecise for $\mathcal{M}$. For measuring the error of $\mathcal{M}_d$ with respect to $\mathcal{M}$, the parameters $\eta_A$ and $\eta_R$ are defined, which intuitively are the number of elements that have to be added to/removed from a maximum-weight basis of $\mathcal{M}_d$ to reach a maximum-weight basis of $\mathcal{M}$.
The main result of the paper is an algorithm that computes a maximum-weight basis of $\mathcal{M}$ using at most $\min (n-r+k+\eta_A \cdot (k+1)+ \eta_R \cdot (k+1) \lceil \log_2 r_d \rceil, (1+1/k)n )$ calls to the clean oracle, where $n=|E|$, $k$ is a positive integer, and $r$ and $r_d$ are the ranks of $\mathcal{M}$ and $\mathcal{M}_d$, respectively. The authors also prove lower bounds that show any deterministic algorithm should have dependencies with respect to $n$, $r$, $\eta_A$, and $\eta_B$ that are similar to those of the proposed algorithm.

**Strengths:**

* The authors provide interesting and nontrivial upper and lower bounds for a fundamental problem.
* The two-oracle model considered, which can be viewed as a learning-augmented model, is theoretically interesting.
* The paper is well-written, and the algorithms are presented in a logical sequence that is easy to follow. The warm-up algorithms introduced in section 2 are especially helpful in facilitating understanding of the techniques and challenges.

**Weaknesses:**

* The parameter $k$ in the main result of the paper is intuitively used to determine how much we want to trust the dirty oracle. When
$k$ is set close to 0 to heavily favor the dirty oracle, if the errors $\eta_A$ and $\eta_R$ turn out to be high, the algorithm might use $\Omega(n \log n)$ calls to the clean oracle. This significantly exceeds the calls required by the optimal worst-case algorithm without predictions, which uses $n$ calls to the clean oracle. In this sense, the algorithm is not robust.
* The results lack empirical verification. Even basic proof-of-concept experiments would be valuable to assess whether this model could be practically applicable.

**Questions:**

* Minor remarks:
  * In lines 71-73 and 86-87, the wording was initially confusing to me. I think the way these results are stated later as the minimum or maximum of two values is easier to read.
  * The error measures defined in lines 375-380 can be exponential in $n$.

**Limitations:**

The authors state their theoretical results formally, describing all assumptions.

---

> ### Author Rebuttal · Authors · 2024-08-07
>
> ### Question Regarding Worst-case Guarantee (claimed n log n bound)
>
> Our wording in lines 71-73 might have been misleading and we will change it. Our robustness guarantee, and thus the overall guarantee of the algorithm, is at most 2n (when k=1, which is the smallest value of k for which the stated guarantee holds). In a revision, we will use the standard minimum expression for combining the error-dependent guarantee and the robustness guarantee.
>
> ### Question/comment about the error measures defined in lines 375-380 to be potentially exponential in n
>
> Yes, this is correct. If this is undesirable, we can adapt it to the number of "wrong" edges (at most rn) in the exchange graph constructed to find the alternating paths, which can be defined as follows: taking the max over all independent sets X, how many pairs (x,y) with X-x+y are dependent but judged as independent by the dirty oracle.
> With this new definition, we can obtain the same results with only a minor modification in the analysis.
>
> ### Empirical Verification
>
> We are happy to add experiments in a full version. Our work provides theoretical results with proofs that hold true without experimental verification. It has become increasingly common in the field of learning-augmented algorithm design to omit experimental verification for purely theoretical results, even when publishing at AI venues.

---

> > ### Comment · Reviewer_KsE6 · 2024-08-12
> >
> > Thanks to the authors for their response. Regarding the robustness guarantees, I mistakenly assumed that $k$ can be arbitrarily close to zero. Thanks for the clarification.
> >
> > After reading all the reviews and responses, I have decided to maintain my original score.

---

### Official Review · Reviewer_2QVw · 2024-07-13

**Soundness:** 3
**Presentation:** 2
**Contribution:** 3
**Rating:** 4
**Confidence:** 4

**Summary:**

The paper explores the concept of 2-oracle algorithms for matroid optimisation problems. The underlying idea is to equip algorithms with a second, somewhat "weaker" oracle. The second oracle is also permits the algorithm to query matroid information (similar to the first oracle) but only gets imprecise answer (different from the first oracle). Therefore the second oracle is also called "fast" as the assumption is that its use is cheaper than the use of the first oracle. In this context the problem of computing a maximum-weight basis for a matroid is investigated. The main result obtained is the existence of an algorithm for computing a maximum-weight basis with a prescribed number of oracle calls. Some related tools and aspects like error-depdent and robust algorithms are also considered.

**Strengths:**

- The main results are established in form of formal theorems that come with a proofs; I did not check the proofs in detail but they look solid (the proof techniques look standard and adequate for this purpose); thus the soundness of the material appears good

- While this is not the first paper that studies the power of 2-oracle algorithms its study in the context of matroid optimisation appears to be original; thus the novelty of the paper is good

**Weaknesses:**

- The paper is not easy to read; at the end of the introduction (i.e. Section 1) there should be an outline of the organisation of the paper to give the reader orientation

- The organisation of the paper is poor: the 9 pages text consist of 4 pages introduction (Section 1), 1.5 pages warm-up (computing an "unweighted" basis) (Section 2), 2 pages computing a max-weighted basis (Section 3) and 1.5 pages future work (Section 4); the way how the material is organised is really unfriendly for the reader

- The main results require a long list of lemmas (which are given in the paper or its appendix); so this might be tedious for the reader to follow the line of proof

- The presentation of the material in the paper is at most fair; changing this may require quite some rewriting of the paper

**Questions:**

- While matroid optimisation is closely related to combinatorial optimisation it is not clear why this paper is submitted to NeurIPS as the connection to neural computing is not that obvious; so this should be justified better

- The paper speaks of "robust" algorithms and of "robustifying" error-dependent algorithms in a sense of dealing with errors from the second oracle; but robust optimisation has been studied widely in the combinatorial optimisation community is usually concerned with uncertainty in the costs or times given as input; so these two meaning of "robust" may easily be mixed up and may be confusing for the reader

---

> ### Author Rebuttal · Authors · 2024-08-07
>
> ### Justification for Conference Fit
>
> Among the major topics of NeurIPS, our paper fits very well under "Optimization" and "Machine Learning".
> While we do not develop new ML methods, we instead design potential applications for ML-learned information. We theoretically analyze their potential for classic optimization problems in the context of learning-augmented algorithm design, which has been quite successfully established in AI venues, including NeurIPS (examples: Bai and Coester, 2023; Purohit, Svitkina and Kumar, 2018) and ICML (examples: Choo et al., 2023; Lykouris and Vassilvitskii, 2018), over the last five to eight years.
> We refer to reference [26] in our submission for an almost complete list of papers in this area, many of which are published at NeurIPS or AI conferences in general.
> In fact, an important use case of this area is to guide the development of ML methods: Which information should ML methods learn in order to improve widely used optimization algorithms?
>
> 1. *Bai, X., and Coester, C. "Sorting with predictions", NeurIPS 2023*
> 2. *Choo, D., et al. "Active causal structure learning with advice", ICML 2023*
> 3. *Purohit, M., Svitkina, Z., and Kumar, R., "Improving Online Algorithms via ML Predictions" NeurIPS 2019*
> 4. *Lykouris, T., and Vassilvtiskii, S. "Competitive Caching with Machine Learned Advice", ICML 2018*
>
> ### Term "robust" and its use in robust optimization
>
> Indeed, robust optimization is a classical field of optimization concerned with worst-case guarantees in the face of uncertainty in the input, either in the objective function (cost) or constraints.
> In the field of learning-augmented algorithms this phrase has been adopted to explicitly refer to worst-case guarantees with respect to the uncertainty about the predictions, which are part of the input.
>
>
> ### Weak presentation
>
> We regret that the reviewer finds the paper difficult to read.
> The page limit imposes serious constraints on presenting the technical material,  but a full version will allow us to provide more details and illustrations.
> As is common in theoretical papers, we aimed to present our technical results in a clear manner by breaking complex proofs into several lemmas, allowing readers to maintain an overview without needing to delve into all the details in the main body of the paper.
> We are unclear about the specific concerns regarding the paper’s organization.
> We hope that any issues with the organization of the sections do not significantly impact the evaluation of the scientific merits of the paper.
> We will add an overview at the end of Section 1 as requested.

---

> > ### Comment · Reviewer_2QVw · 2024-08-13
> >
> > Dear authors,
> > Many thanks for your answers that helped in clarifying several points. I am more happy now.
> >
> > Regarding 1) Justification for Conference Fit. The motivation for submitting the paper is clearer now. The connection to learning-augmented algorithms could be explained a bit more detailed (as done here in your reply). In my opinion, reference [26] is not a solid argument as it is unclear according to which criteria this list of papers in Github is put together and maintained. Perhaps you can phrase this differently.
> >
> > Regarding 2) Term "robust" and its use in robust optimization. Thanks for the explanation. You could also include such a brief remark into the paper to avoid confusion.
> >
> > Regarding 3) Weak presentation. I understand that the page limit is a challenge, and you made a good effort to put a lot of content into the available space. My concern is not that you break a complex proof into a series of lemmas. I also like that you first study the unweighted case before you continue with the more general weighted case. My concern is rather that many readers will give up before they reach the end of the introduction (at the bottom of page 4), and that would be a pity. Good that you offered to include an overview of the paper’s organisation at the end of the introduction. The future reader of your paper will definitely benefit from this. (My recommendation would even be to subdivide the 4-page introduction and place this overview earlier; but this is up to you.)

---

### Decision · Program_Chairs · 2024-09-25

**Decision:**

Accept (poster)

**Comment:**

The reviewers remark that claims appear to be well substantiated with proofs [2QVw, KsE6], the manuscript presents strong results [rC5n, AWmg] and is well-written/clear/logically [KsE6, AWmg], but the writing/organization/readability of the paper could still be improved [2QVw, rC5n], the complexity improvement for independence queries hinge on the rank of the matroid r to grow with larger element cardinality $n$ such that $n-r$ would grow sublinearly [AWmg], limited improvement over trivially running two algorithms in parallel [AWmg] and requiring strong assumptions for the extension to matroid intersection [AWmg].

Apart from answering reviewer questions, the rebuttal expresses the intention to add a full version, not bound by page limits, which features more details and illustrations [response to 2QVw], a gentle introduction to matroid optimization [response to rC5n], some empirical verification [responses to KsE6, rC5n], clarifications of how $2$-oracle matroid optimization relates to neural networks [response to 2QVw] and that "robust" refers in this context to worst-case guarantees over inputs [response to 2QVw].

The overall recommendation is to accept this submission with the agreed upon changes during the discussion phase. While the improvements might not be particularly substantial, there is not really much room for improvement as the bounds are tight [AWmg], the work comprehensively studies a relevant problem and the proofs as well as presentation appear to be generally of high quality although the writing might be a bit dense at times.